# Optical tweezers-controlled hotspot for sensitive and reproducible surface-enhanced Raman spectroscopy characterization of native protein structures

Xin Dai[1,2], Wenhao Fu[1], Huanyu Chi[1], Vince St. Dollente Mesias[1], Hongni Zhu[1], Cheuk Wai Leung [1], Wei Liu[3✉] & Jinqing Huang [1✉]

Surface-enhanced Raman spectroscopy (SERS) has emerged as a powerful tool to detect biomolecules in aqueous environments. However, it is challenging to identify protein structures at low concentrations, especially for the proteins existing in an equilibrium mixture of various conformations. Here, we develop an in situ optical tweezers-coupled Raman spectroscopy to visualize and control the hotspot between two Ag nanoparticle-coated silica beads, generating tunable and reproducible SERS enhancements with single-molecule level sensitivity. This dynamic SERS detection window is placed in a microfluidic flow chamber to detect the passing-by proteins, which precisely characterizes the structures of three globular proteins without perturbation to their native states. Moreover, it directly identifies the structural features of the transient species of alpha-synuclein among its predominant monomers at physiological concentration of 1 μM by reducing the ensemble averaging. Hence, this SERS platform holds the promise to resolve the structural details of dynamic, heterogeneous, and complex biological systems.

---

[1] Department of Chemistry, The Hong Kong University of Science and Technology, Clear Water Bay, Hong Kong, China. [2] Laboratory for Synthetic Chemistry and Chemical Biology, Health@InnoHK, Hong Kong Science Park, Hong Kong, China. [3] State Key Laboratory of Synthetic Chemistry, Department of Chemistry, The University of Hong Kong, Pokfulam Road, Hong Kong, China. ✉email: wliu276@hku.hk; jqhuang@ust.hk

Raman spectroscopy probes the endogenous vibrations of molecules upon irradiation to delineate their chemical structures and surrounding environments[1], feasible for the label-free characterization of biomolecules in aqueous environments[2,3]. Inheriting these advantages, surface-enhanced Raman spectroscopy (SERS) boosts up the sensitivity to detect proteins at low concentrations, even at the single-molecule level, to mimic physiological conditions[4–6]. It is also a powerful tool to characterize the dynamic ensembles of variable conformations of intrinsically disordered proteins (IDPs)[7–10]. IDPs lack stable secondary and tertiary structures as monomers in aqueous environments, but sometimes self-assemble into oligomers with various structures and further grow into amyloid fibrils[11], which are associated with the incurable neurodegenerative diseases[12]. The dynamic conversion from monomers to oligomers is the key step in the early pathological development[13]. However, the transient nature and the low population of oligomers make it challenging to characterize their structural features, which are responsible for the cellular toxicity and the on-going amyloid aggregation[14]. In particular, the SERS study on alpha-synuclein, an IDP closely related to Parkinson's disease[15–17], is scarce[18]. Besides, practical difficulties emerge when using conventional nanoparticle-based SERS substrates generated from salt-induced random aggregations to probe proteins in dilute solutions, such as low efficiency, poor reproducibility, and potential structural perturbation during long incubation time in SERS measurements[19,20]. Hence, there is an urgent need to customize the SERS approach to better characterize protein structures at low concentration for greater biological significance.

The SERS activity arises from the induced strong electromagnetic field confined in the junctions between plasmonic nanostructures, known as "hotspots"[21,22]. The salt-induced aggregation of silver nanoparticle (AgNP) colloids is the most accessible method to form SERS active gaps[23–25]. Yet such random aggregates lead to unequal enhancement factors and fluctuating SERS signals[26]. Thus, one major challenge in the field is to improve the reproducibility. Advancements have been achieved mainly on fabricating SERS substrates with sharp size

distribution[27–29], in order to provide consistent SERS enhancements. Nevertheless, when measuring analytes at low concentrations, it is inconvenient to locate and focus on the nanometer-size hotspots from the conventional aggregated nanoparticles in colloidal suspension for in situ detection, thus lowering the feasibility and efficiency of the technique. To address this dilemma, mechanical manipulations are desirable alternatives to create hotspots dynamically at the designated location[7,30,31]. Recently, optical tweezers-assisted SERS has been developed to increase the efficiency for the SERS analysis in aqueous conditions[32–41], by introducing the additional trapping laser beams to manipulate SERS substrates[42,43]. Kall and co-workers utilized optical tweezers to bring two AgNPs together to form a SERS active dimer[44], but the hotspot size and the precise location was uncertain due to the diffraction limit of optical microscope[44]. Smith and co-workers trapped one partially Ag-coated silica microparticle as a mobile and visible SERS probe[45], which was further developed by Petrov and co-workers to accomplish interfacial detection of living cells[46]. However, there is still lack of an explicit control on the hotspot and the subsequent SERS performance. The optical tweezers-based SERS platform awaits further developments to exploit its mechanical control to improve the efficiency, sensitivity, and reproducibility for the investigation of various biomolecules in dilute environments.

In this work, we introduce a convenient approach to visualize and control hotspots to provide consistently high SERS enhancements for the characterization of protein structures and conformational fluctuations at physiological concentration, by developing the optical tweezers-coupled Raman spectroscopy. Specifically, two AgNP-coated micrometer-size silica beads are trapped and approached to form the SERS active interparticle gap under the precise manipulation and real-time visualization for in situ spectroscopic measurements, even offering single-molecule level sensitivity. This dynamic and tunable SERS window is placed in a microfluidic flow chamber to detect the passing-by proteins. The flow rate of the protein solution is fine-tuned to minimize the interaction between proteins and AgNP-coated beads, in order to preserve their native states and conformations.

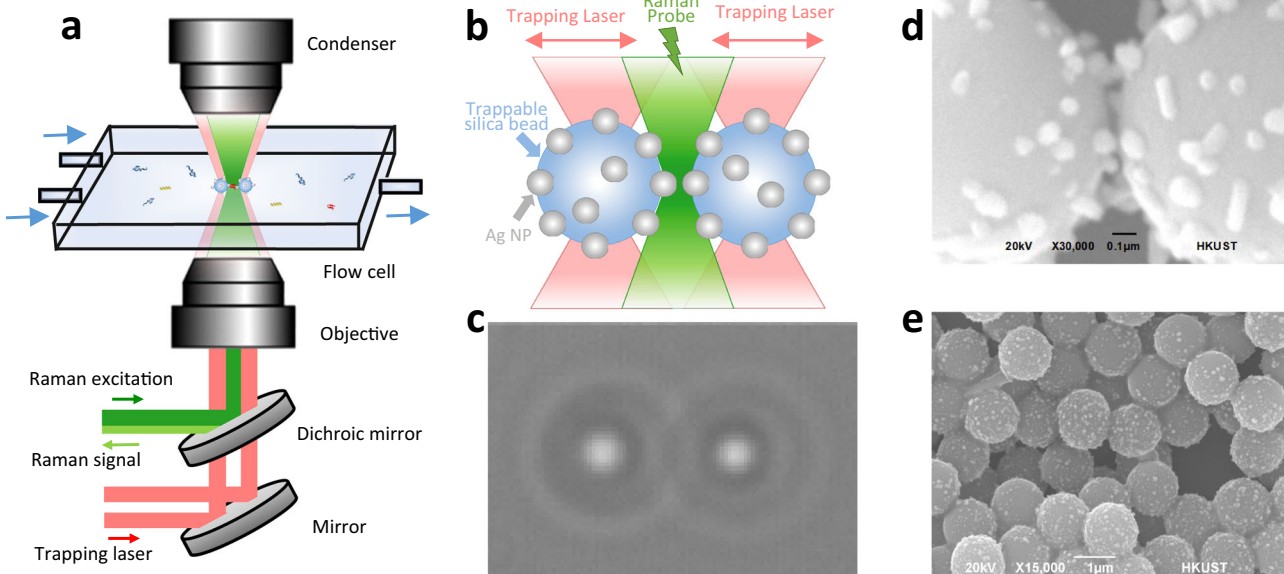

**Fig. 1 Illustration of the controllable SERS probe experiments. a** Schematic diagram of the optical tweezers-coupled Raman spectroscopic platform with microfluidic set-up. **b** Two trapping laser beams (red) to manipulate two AgNP-coated beads and one Raman probe beam (green) to detect signals from the gap between the two AgNP-coated beads. **c** The real-time camera image of two trapped AgNP-coated beads from the microscope. **d** SEM image of the gap between two AgNP-coated beads. The scale bar is 0.1 μm. **e** SEM image of AgNP-coated beads to show the uniform AgNP coating. The scale bar is 1 μm. All micrographs are representative images of three independent measurements.

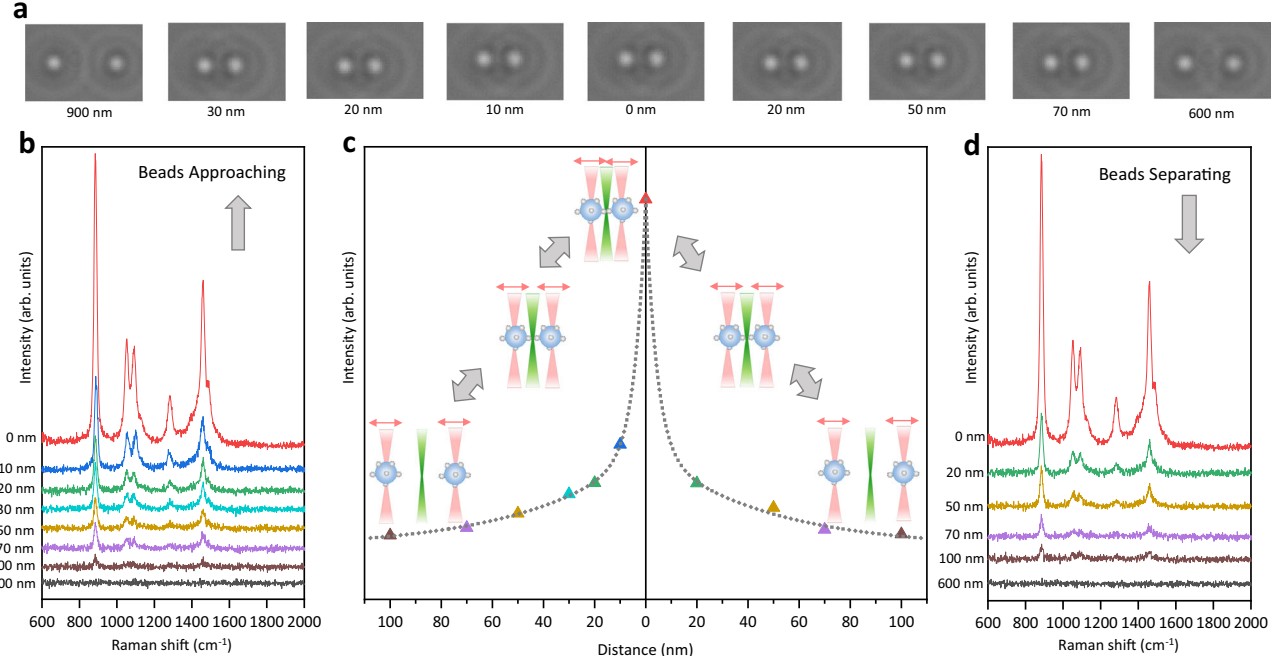

**Fig. 2 Creation and adjustment of the dynamic hotspot with in situ SERS measurements of 1% ethanol aqueous solution. a** The real-time camera images of the two AgNP-coated beads trapped at different distance. **b** SERS spectra of 1% ethanol aqueous solution with 1 s acquisition time when the two AgNP-coated beads approaching. **c** The intensity of the ethanol characteristic peak at 1458 cm$^{-1}$ as a function of the distance between the two AgNP-coated beads from beads approaching to beads separating. The reversible bead positions are illustrated as inset. **d** SERS spectra of 1% ethanol aqueous solution with 1 s acquisition time when the two AgNP-coated beads separating. Source data are provided as a Source data file.

The structural features of three typical globular proteins (hemoglobin, lysozyme, and bovine serum albumin) and an IDP (alpha-synuclein) are analyzed in dilute solutions. In particular, the SERS characterization of alpha-synuclein at physiological concentration (1 μM)[47] is reported, showing the structural variations arisen from its transient species. This sensitive, reliable, and convenient SERS platform enables flash spectroscopic snapshots of proteins at low concentration to reduce the ensemble averaging in time and in quantity, thus has great potential to investigate IDP oligomers to provide perspectives on the initiation of amyloid protein aggregation and tackle the problems in complex biological systems.

## Results

### The optical tweezers-coupled Raman microscope and the trappable SERS substrate.
The home-built SERS platform combines dual-trap optical tweezers and Raman microscope. As illustrated in Fig. 1a, two 1064 nm trapping laser beams (red) and one 532 nm Raman probe beam (green) propagate inside the inverted microscope through stereo double-layer-pathways, which are then combined by a dichroic mirror to enter the objective from the bottom. Spatially separated, the beams are focused inside a flow chamber connected to the microfluidic system. The two trapping laser beams manipulate the flowing AgNP-coated beads in 3D while the Raman probe beam irradiates the gap between two trapped AgNP-coated beads as shown in Fig. 1b. The backscattered light is collected for spectroscopic measurements. In the forward direction, the two trapping laser beams are collimated by a condenser for force detections and camera imaging. (More instrumental details are illustrated in Supplementary Fig. 1). Figure 1c shows the real-time camera image of the two trapped AgNP-coated beads for direct distance adjustments. This instrumental set-up enables the optical manipulation of SERS substrates under microscopic visualization and the in situ SERS measurements at the same time.

The AgNP-coated beads were chemically fabricated for manipulation and visualization as the dynamic SERS probe[45,46,48]. First, we modified the surface of micrometer-size silica beads with amino groups in anhydrous ethanol, then coated them with AgNP colloids in continuous agitation. The AgNPs were prepared from the reduction of silver nitrate by trisodium citrate and washed three times to remove the reducing agent before mixing with the modified silica beads. The parameters such as interaction time, stirring rate, beads size, and the reactant ratio were fine-tuned to generate the uniform AgNP coating at the density of 17.6% on the beads (Fig. 1e), in order to ensure the reproducible and stable optical trapping performance. (Details are exhibited in Supplementary Fig. 2.) Figure 1d displays the SEM image of the interparticle gap when two AgNP-coated beads are in close proximity, which could determine the SERS activity. Compared to the direct trapping of AgNPs, the development of micrometer-size AgNP-coated beads enables the precise position control with sub-nanometer spatial resolution while minimizes their Brownian motions for more effective, reproducible, and stable spectroscopic measurements. (Details are demonstrated in Supplementary Figs. 3 and 4.) Moreover, the visualization of the SERS hotspot between the AgNP-coated beads on the brightfield camera or the eyepiece of Raman microscopes makes it easy to adjust the Raman excitation focus, superior to the dispersed and invisible AgNP colloids with non-specific aggregations in dilute solutions. Thus, the strategy to create the hotspots between micrometer-size AgNP-coated beads improves the operational efficiency and simplicity for sensitive SERS detections.

### The creation of the real-time controllable hotspot.
As a proof of concept, the dynamic SERS probe was created and adjusted between the two AgNP-coated beads on the optical tweezers-coupled Raman microscope to detect the SERS signal of 1% ethanol aqueous solution. Two AgNP-coated beads were trapped at different distance reversibly, meanwhile the real-time camera

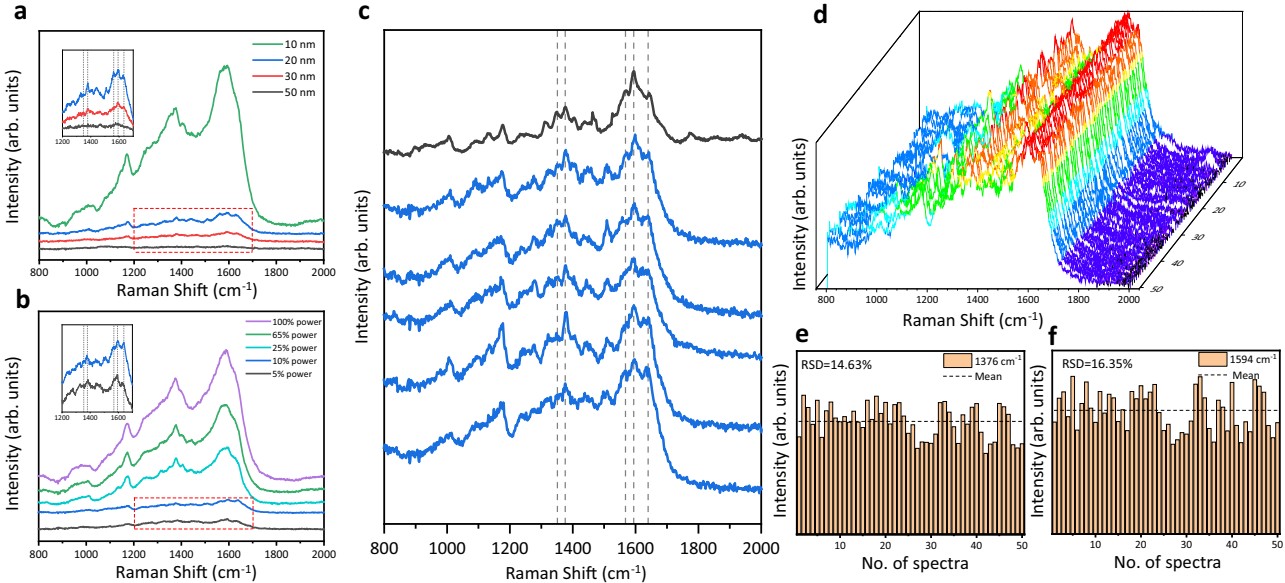

**Fig. 3 The spectroscopic characterizations of hemoglobin at its native states. a** SERS spectra of 100 nM hemoglobin in aqueous solution when the two AgNP-coated beads were trapped at different distance (50–10 nm). **b** SERS spectra of 100 nM hemoglobin solution under different Raman excitation power (5–100%). **c** The comparison between SERS spectra of 100 nM hemoglobin solution with 1 s acquisition time (blue) and the spontaneous Raman spectrum of 250 μM hemoglobin solution with 60 s acquisition time (black). **d** 3D stacking plot of SERS spectra of 100 nM hemoglobin solution obtained from AgNP-coated beads trapped at 20 nm with 1 s acquisition time. **e, f** The histograms of the intensities of the hemoglobin characteristic peaks at 1376 cm$^{-1}$ (mean = 4929.15 in arb. units, RSD = 14.63%) and 1594 cm$^{-1}$ (mean = 5391.41 in arb. units, RSD = 16.35%) across the 50 SERS spectra in (**d**), respectively. Source data are provided as a Source data file.

images (Fig. 2a) and the corresponding SERS spectra with 1 s acquisition time (Figs. 2b and d) were recorded. At the beginning, the two beads were separated away by 900 nm on the two sides of the Raman excitation spot to acquire the blank spectrum in Fig. 2b (black), which is the same as spontaneous Raman of 1% ethanol in Supplementary Fig. 5 showing no apparent signal. The two beads were approached to decrease the distance below 100 nm, while the Raman signal gradually emerged and enhanced, indicating the formation of the SERS active gap at the Raman excitation spot. The characteristic peaks at 886 cm$^{-1}$ (C–C stretching), 1054 cm$^{-1}$ (C–O stretching), 1094 cm$^{-1}$ (CH$_3$ rocking), 1280 cm$^{-1}$ (CH$_2$ deformation) and 1458 cm$^{-1}$ (CH$_3$ asymmetric bending)[49] are consistent with the spontaneous Raman of pure ethanol shown in Supplementary Fig. 5. To confirm the SERS enhancement effect from the hotspot between the two approached AgNP-coated beads, the entire surface of individual AgNP-coated beads was scanned in time-series measurements. The blank spectra of the single AgNP-coated bead trapped at the Raman excitation spot before and after the hotspot creation in Supplementary Fig. 6 imply no intra-bead hotspots. It is worth noting that the spectral features were boosted up during beads approaching in a 10 nm step-size from 30 nm to 0 nm, indicating that the SERS active interparticle gap was under precise control and the slight adjustment resulted in the dramatic signal enhancement. In the reverse moving direction, the SERS signal gradually decreased when the two AgNP-coated beads were separated further from 20 nm to 100 nm, shown in Fig. 2d. At the same beads distance, the signal intensities obtained in beads separating are comparable to those obtained in beads approaching. Figure 2c demonstrates the intensity of the ethanol characteristic peak at 1458 cm$^{-1}$ as a function of the distance between the two trapped AgNP-coated beads, obtained from the trapping laser position and the bright-field image analysis. It is clear that the vibrational features of ethanol were emerged, enhanced, decreased, and vanished in inversely proportional relationship to the beads distance, under the precise control by optical tweezers with excellent flexibility and

reproducibility. Furthermore, the two trapped beads were pushed at a distance of 0 nm to maximize the SERS enhancement and generate the detectable force between them to confirm the contact of two beads (Supplementary Fig. 7) and ensure the stability of the SERS active interparticle gap (Supplementary Fig. 6). It was utilized to probe 10$^{-9}$ M and 10$^{-11}$ M Rhodamine B, showing the Raman signatures of the aromatic C–C stretching of Rhodamine B at ~1360 cm$^{-1}$, ~1565 cm$^{-1}$, and ~1650 cm$^{-1}$ in Supplementary Fig. 8[50]. The theoretical simulation indicated its SERS enhancement factor up to 10$^9$, which is sufficient to empower the single-molecule level detection.[51] (Details are explained in Supplementary Fig. 8.) Overall, the creation and the adjustment of hotspot could be achieved efficiently and reversibly at the Raman excitation spot, which offers a convenient platform to generate tunable and reproducible SERS enhancements for various analytes.

**The spectroscopic characterizations of native hemoglobin.** We utilized the controllable SERS probe as a detection window to analyze the flowing hemoglobin in dilute aqueous solution without incubation or aggregation with AgNPs. As illustrated in Supplementary Fig. 9, two AgNP-coated beads were trapped in the microfluidic bead channel then moved to the hemoglobin channel for SERS measurements. The flow rate of the hemoglobin solution was fine-tuned to minimize the interaction between proteins and AgNP-coated beads. The spectroscopic scans of the individual AgNP-coated beads at the Raman excitation spot show the blank spectra in Supplementary Fig. 10 to confirm neither intra-bead hotspots existence nor hemoglobin attachments. Under the visualization and manipulation by optical tweezers, the two AgNP-coated beads were approached at incremental distance and gradually adjusted Raman excitation power to optimize the SERS signal of 100 nM hemoglobin in aqueous solution in Fig. 3a and b. When the beads distance was smaller than 20 nm and the laser power was larger than 10% (2.5 mW), the subtle spectral features were overwhelmed by the intense characteristic peaks arisen from

amorphous carbon (~1370 cm$^{-1}$ and ~1580 cm$^{-1}$)[52], suggesting molecular damages. Thus, the SERS detection window was set under the bead distance at 20 nm and the Raman excitation power at 10% with 1 s acquisition time. It is showcasing the great adjustability of the optical tweezers-coupled Raman spectroscopy to preserve the protein native states.

To examine if the hemoglobin retains native, five SERS spectra of 100 nM hemoglobin with 1 s acquisition time (blue) were compared to the spontaneous Raman spectrum of 250 μM hemoglobin with 60 s acquisition time (black) in Fig. 3c. The identical frequencies between the SERS spectra and the Raman spectrum of hemoglobin indicates the similar protein states in measurements[3]. As the oxidation state marker bands, the subtle peaks at 1346 cm$^{-1}$ and 1376 cm$^{-1}$ among these spectra are distinctly similar, attributed to the ferrous state and the ferric state[53], respectively. Vibrational bands at 1568 cm$^{-1}$ and 1594 cm$^{-1}$ corresponding to 6-coordinated high-spin heme and 6-coordinated low-spin heme are apparent at matching positions[53,54]. Bands appearing at 1089 cm$^{-1}$, 1311 cm$^{-1}$ and 1440 cm$^{-1}$, assigned to the vinyl group deformation in the porphyrin ring of the heme center, are in good agreements.[55] Whereas, the marker bands of 5-coordinated high-spin heme at 1494 cm$^{-1}$ and 1572 cm$^{-1}$[53,54], representing the non-native state from the perturbation of metal surface, were not observed. All peak assignments of hemoglobin are listed in Supplementary Table 1[53,55,56–58]. It is clear that the SERS probe created by two AgNP-coated beads well preserved the oxidation state, the coordination number, and the spin state of hemes in hemoglobin, which are closely linked to the native structure and function of hemoglobin[54]. Furthermore, the high similarity of these SERS spectra demonstrates the reproducibility of this SERS platform.

To further preserve the protein native states, the two AgNP-coated beads were replaced freshly in parallel SERS measurements to minimize the interaction time with proteins[19,59]. Figure 3d displays SERS spectra of 100 nM hemoglobin solution obtained from 50 parallel measurements. Apparently, the vibrational signatures of the heme center of hemoglobin at 1346, 1376, 1568, and 1594 cm$^{-1}$ among the 50 SERS spectra are approximately the same. Figure 3e and f demonstrate the histograms of the peak intensities at 1376 cm$^{-1}$ and 1594 cm$^{-1}$ across the 50 SERS spectra with relative standard deviation (RSD) as 14.63% and 16.35%, respectively. Histogram analysis of other spectral features is presented in Supplementary Fig. 11. These reproducible and stable spectra prove the consistent SERS enhancements in the parallel measurements when two AgNP-coated beads were trapped at a constant distance. Hence, the controllable SERS probe inside the microfluidic flow chamber could preserve the flowing proteins at their native states and generate reproducible SERS spectra.

**The spectroscopic characterizations of the compact globular structure of lysozyme**. To demonstrate the ability to detect protein conformations with high accuracy, we employed the dynamic SERS probe as a detection window to characterize two well-known globular proteins in solutions: lysozyme and bovine serum albumin (BSA). With the experimental protocol analogous to the previous section, two fresh AgNP-coated beads were trapped at 20 nm to analyze the flowing proteins in dilute aqueous solutions, which would be replaced freshly in parallel experiments. Figure 4a displays the SERS spectra of 1 μM lysozyme solution acquired from 50 parallel experiments, showing the vibrational frequencies of lysozyme identical to its spontaneous Raman spectrum. Specifically, the peaks at 766, 1015, 1337, and 1557 cm$^{-1}$ are assigned to the aromatic residues (tryptophan) and the peak at 1450 cm$^{-1}$ is attributed to aliphatic residues

(CH$_2$)[60]. The amide I band at 1655 cm$^{-1}$ and the amide III band at 1250 cm$^{-1}$ imply the existence of α-helix in the folded globular structure of lysozyme. The spectral contributions from different secondary structures were deconvolved to α-helix (45.2%), β-sheet (11.3%), and random coil (43.5%) in Supplementary Fig. 12, which is consistent with the previous investigations[61,62]. Figure 4b demonstrates the Amide I band distribution of the 50 SERS spectra of 1 μM lysozyme solution with 1655 ± 2 cm$^{-1}$ (0.1% RSD), indicating its structural stability and homogeneity as a typical globular protein. Moreover, the SERS spectra of 1 μM BSA in Supplementary Figs. 13 and 14 also provide the structural component assessment (66.5% α-helix, 9.5% β-sheet and 24.1% random coil) supported by the previous studies[63]. It is worth noting that the 50 SERS spectra of 1 μM lysozyme in Fig. 4a demonstrate high similarities, due to the stability and reproducibility of this SERS probe as well as the nature of the stable, compact, and globular conformation of lysozyme. Since the sizes of these globular proteins are smaller than the bead distance at 20 nm[62], these SERS spectra reflect the whole protein structures. As illustrated in Fig. 4c, the small-size sampling in the parallel SERS measurements unveil the protein structural fluctuation to complement the ensemble averaging in the spontaneous Raman measurement.

**The spectroscopic investigation of the transient species of alpha-synuclein at physiological concentration**. To exploit the advantages of the controllable SERS probe, we characterized the structural features of alpha-synuclein in aqueous solutions. As a typical IDP, alpha-synuclein undergoes intrinsic conformational conversions to form transient species in a low population at physiological concentration, existing in a dynamic equilibrium mixture to determine its amyloid aggregation at the early stage[64]. The CD spectrum of 200 μM alpha-synuclein aqueous solution in Fig. 5a presents a negative peak at around 200 nm, indicating the ensemble conformations as random coils[13]. In Fig. 5b, the spontaneous Raman spectrum of 2 mM alpha-synuclein solution shows the vibrational features in amide III region at 1249 cm$^{-1}$ and amide I region at 1673 cm$^{-1}$, which are attributed to the majority population of disordered conformations[16]. The protein signal of 250 μM alpha-synuclein solution in Fig. 5b is too weak to analyze, due to its low Raman cross-section and the gentler laser power (25 mW) in our setup, lower than the 800 mW laser power used in the previous Raman studies[16,17]. With the large sample quantity and long detection time in the bulk spectroscopic measurements, the structural features of alpha-synuclein transient species are overwhelmed by the conformational ensemble.

The physiological concentration of alpha-synuclein at non-aggregated states is 1 μM[47], below the detection threshold of Raman spectroscopy. Whereas the experimental limit of detection (LOD) of alpha-synuclein on our SERS platform is 100 nM, thus this sensitive SERS approach is feasible to characterize the transient species of alpha-synuclein in dilute solutions. (Details are shown in Supplementary Figs. 15 and 20.) Similar to the experimental protocol in the previous section, two fresh AgNP-coated beads were trapped at 20 nm as the dynamic SERS window to characterize the flowing alpha-synuclein on our platform. Figure 5c demonstrates three representative types of SERS spectra of 1 μM alpha-synuclein solution with 1 s acquisition time among the 200 parallel experiments shown in Fig. 5d. Strikingly, the amide I and the amide III bands of the SERS spectra of 1 μM alpha-synuclein exhibit prominent variations in comparison to the uniform spectral patterns of 1 μM lysozyme in Fig. 4, indicating the co-existence of different alpha-synuclein species with various structures at physiological concentration[7]. In Fig. 5c, the spectral characteristics of the blue SERS spectrum of 1 μM alpha-synuclein fall in the

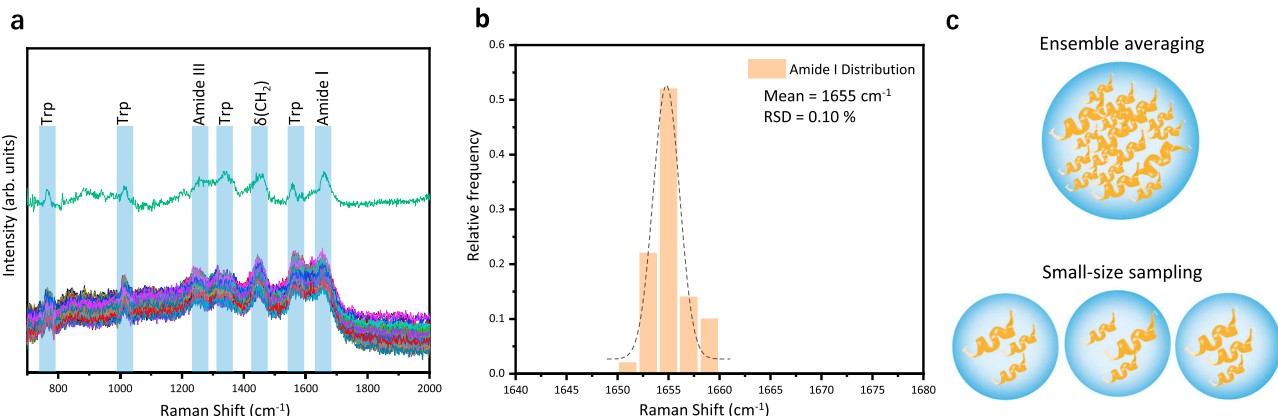

**Fig. 4 The spectroscopic characterizations of lysozyme in its compact globular structure. a** The comparison between 50 SERS spectra of 1 µM lysozyme solution with 1 s acquisition time (bottom) and the spontaneous Raman spectrum of 1 mM lysozyme solution with 5 min acquisition time (top). **b** Histogram of the Amide I band distribution of the 50 SERS spectra of 1 µM lysozyme solution in (**a**), indicating the mean as 1655 cm$^{-1}$ with 0.1% RSD. **c** Illustration of the ensemble averaging from the spontaneous Raman measurement of lysozyme in the concentrated solution and the small-size sampling from the SERS measurements of lysozyme in the dilute solution. Source data are provided as a Source data file.

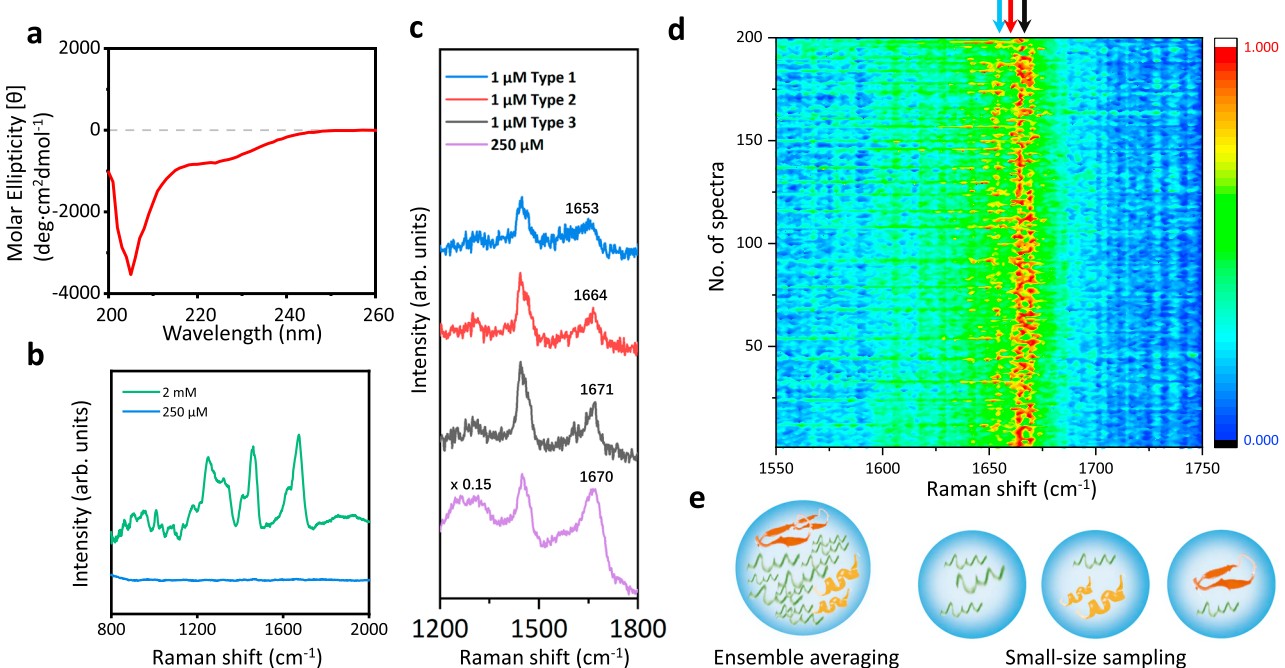

**Fig. 5 The spectroscopic characterizations of an intrinsically disordered protein: alpha-synuclein. a** CD spectrum of 200 µM alpha-synuclein in aqueous solution. **b** Spontaneous Raman spectra of 2 mM (green) and 250 µM (blue) alpha-synuclein solution with 10 min acquisition time. **c** The comparison among three representative types of SERS spectra of 1 µM alpha-synuclein solution with 1 s acquisition time (blue, red, and black) and the SERS spectrum of 250 µM alpha-synuclein solution with 5 min acquisition time (purple). **d** Mapping of 200 SERS spectra of 1 µM alpha-synuclein solution obtained from two AgNP-coated beads trapped at 20 nm with 1 s acquisition time. The color bar shows the normalized intensities from low (dark blue) to high (red). **e** Illustration of the ensemble averaging from the measurement of alpha-synuclein at high concentration with long accumulation time and the small-size sampling from the measurements of alpha-synuclein at low concentration with short accumulation time. Source data are provided as a Source data file.

intervals of 1650–1657 cm$^{-1}$ (amide I bands) and 1270–1300 cm$^{-1}$ (amide III bands), indicating the alpha-synuclein species in α-helix structure[16,65]. The red SERS spectrum of 1 µM alpha-synuclein shows the amide I band at around 1662–1665 cm$^{-1}$ and the amide III band at 1230–1240 cm$^{-1}$, which are associated with β-sheet structure[16,17,65] This assignment is further confirmed by the SERS spectrum of 1 µM alpha-synuclein at pH 3 in Supplementary Fig. 16, since alpha-synuclein folds into ordered β-sheet conformations in acidic conditions. The amide I band at around 1671 cm$^{-1}$ and the amide III band at 1240–1250 cm$^{-1}$ from the black SERS

spectrum of 1 µM alpha-synuclein are attributed to random coil structure[16,65,66], consistent with its spontaneous Raman spectrum. While the vibrational fingerprints from Phe (1006 cm$^{-1}$) and deformation from aliphatic residues CH$_2$ and CH$_3$ (1450 cm$^{-1}$) of alpha-synuclein[16] are still uniform across these SERS spectra of 1 µM alpha-synuclein, since they are insensitive to the change of protein conformations. All the peak assignments of the 1 µM alpha-synuclein SERS spectra are summarized in Supplementary Table 2[16], since the subtle spectral features of alpha-synuclein acquired at physiological concentration could reveal the structural details of its

transient species with great biological significance. In particular, the direct characterization of the β-sheet containing oligomers among the unstructured monomers of alpha-synuclein might provide further insight to the pathological aggregation of alpha-synuclein at the very early stage as this conformation is involved prior to fibrillation[67,68].

Owing to high sensitivity and stability, our SERS platform resolved the structural variations of alpha-synuclein arisen from its transient species at physiological concentration. In the statistics of the 200 parallel SERS measurements of 1 μM alpha-synuclein solution in Fig. 5d, the probability to observe monomers in random coil structures was very high, since they are the predominant species. Occasionally, the transient species showing α-helix or β-sheet structures were observed. It is consistent with the spectral deconvolution of the 2 mM alpha-synuclein Raman spectrum which shows 13.3% α-helix, 8.4% β-sheet and 78.3% random coil in Supplementary Fig. 18 and the 250 μM alpha-synuclein SERS spectrum which presents 15.1% α-helix, 9.5% β-sheet and 75.4% random coil in Supplementary Fig. 19. The overall population of alpha-synuclein transient species is low at physiological concentration, resulting in a lag phase prior to the growth of fibrils at macroscopic-level. As illustrated in Fig. 5e, the dynamic SERS probe enables the small-size sampling in the measurements of alpha-synuclein in low concentration solution with short accumulation time to directly characterize the structural features of different alpha-synuclein species. The statistics of the structural variations of alpha-synuclein from its transient species provides the distribution of different secondary structure components, comparable and complementary to the ensemble averaging in the measurement of alpha-synuclein in high concentration solution with long accumulation time. Such direct identification of the structural variation of alpha-synuclein verifies that our sensitive SERS platform could reduce the ensemble averaging to reveal more structural information on IDP transient species, providing perspective to investigate the behaviors and functions of IDPs during complex biological processes.

## Discussion

In summary, we mechanically controlled a dynamic SERS probe to characterize typical globular proteins and intrinsically disordered proteins in dilute solutions, using in situ optical tweezers-coupled Raman spectroscopy. Under microscopic visualization and precise manipulation, two AgNP-coated beads were approached by optical tweezers to create tunable hotspots for efficient, reproducible, and convenient SERS measurements with single-molecule level sensitivity. This dynamic SERS detection window was utilized in the microfluidic flow chamber to detect the flowing proteins at their native states and confirmations, verified by the spectral analysis of hemoglobin, lysozyme, and BSA. With high sensitivity and stability, it resolved the structural variations of alpha-synuclein arisen from its transient species in the low population at physiological concentration, which are buried under the averaging signals in the conventional bulk measurements but crucial for the understanding of the initiation of its amyloid aggregation. Hence, the controllable SERS probe on the optical tweezers-coupled Raman platform is feasible to reveal the unperturbed structural information of various proteins in dilute solutions.

Our strategy enables the precise control of the hotspot between the two trapped micrometer-size AgNP-coated beads to improve the SERS efficiency and reproducibility in the aqueous detections. Except for the tunable SERS enhancement, the coupled optical tweezers also offer sub-nanometer spatial resolution and sub-piconewton force sensitivity to monitor light-matter interactions

in the plasmonic hotspot for extra physical insight. More importantly, our method opens a door to characterize the structural variations of IDPs in dilute solutions, which remains a significant challenge in the biophysics community. This dynamic SERS probe has great potential to investigate the oligomeric state of amyloidogenic proteins and resolve the structural details as well as the conformational population of these transient species prior to the amyloid aggregation, providing profound molecular insights to understand the onset of neurodegenerative diseases. Ultimately, it will be exciting to fully exploit the precise force manipulation of the integrated optical tweezers to unfold a single protein inside the controllable hotspot to resolve its structural dynamics.

## Methods

**Chemicals**. Silica beads were purchased from Spherotech Inc. Silver nitrate (≥99.0%), trisodium citrate (≥99.0%), Tris (≥99.9%), sodium chloride (≥99.0%), (3-aminopropyl) triethoxysilane (≥98.0%), lysozyme (≥95.0%) and bovine serum albumin (≥96.0%) were purchased from Sigma-Aldrich. Yeast extract was purchased from Fisher BioReagents™. Tryptone was purchased from Oxoid™. Hemoglobin was purchased from Worthington Biochemical Corporation.

**Expression and purification of proteins**. Recombinant human wild type alpha-synuclein was overexpressed by E. coli BL21(DE3) with plasmid pET28a. Cells were grown in LB medium in the presence of 50 μg/mL Kanamycin and protein expression was induced by 0.3 mM isopropyl β-D-1-thiogalactopyranoside (IPTG). The cell pellet was resuspended in Tris buffer (25 mM Tris-HCl, pH 7.4) and lysed by sonication. After centrifugation at 30,000 × g for 45 min at 4 °C, the supernatant was boiled to remove most E. coli proteins. After centrifugation at 30,000 × g for another 60 min at 4 °C, the supernatant was loaded onto HiPrep DEAE FF 16/10 column (GE Healthcare). A gradual sodium chloride gradient was chosen and applied to elute the target protein. After SDS-PAGE gel analysis, fractions containing the target protein were desalted by HiPrep™ 26/10 Desalting column. The desalted solution was loaded onto HiPrep 16/60 Sephacryl S-100 column (GE Healthcare) for further purification, with 25 mM Tris-HCl, pH 7.4 buffer as the running buffer. Fractions were analyzed by SDS-PAGE gel, and targeted protein were collected, concentrated and stored at −20 °C. The concentration of alpha-synuclein was determined by UV-1800 spectrophotometer (SHIMADZU) with the extinction coefficient of 5960 cm$^{-1}$ M$^{-1}$ at 276 nm.

**Fabrication of AgNP-coated beads**. A step-by-step protocol describing the fabrication of AgNP-coated beads can be found at Protocol Exchange[69]. AgNP-coated beads were prepared by coating the AgNPs on the surface of silica beads (R = 0.63 μm)[45,46]. First, the silver nanoparticles were synthesized based on the conventional method with optimized parameters[3,50]. 50 mL 1 mM AgNO$_3$ aqueous solution was heated to boiling, followed by the drop by drop addition of 1.0 mL 0.1 M trisodium citrate solution. The mixture was kept boiling for 16 min under constant stirring, then was cooled down to room temperature showing yellowish gray color. The AgNPs colloid was washed with Milli-Q ultrapure water for three times, in order to remove the excess reducing agent. Meanwhile, the surface of the micrometer-size silica beads was modified for the AgNP coating. Silica beads stock (5.0% w/v) was dried at 60 °C before dispersed in anhydrous ethanol. Then, a 0.2% ethanol solution of (3-aminopropyl)triethoxysilane (APTES) was added into beads suspension (final concentration 0.1%) for 24 h shaking incubation at room temperature. The solution was purified by centrifugation with distilled ethanol at 1055 × g for three times to discard the supernatant. After drying at 60 °C to remove the ethanol and adding double-distilled water, the silica beads with the amino groups modified surface were obtained. Lastly, the AgNP-coated beads were prepared in 1 mL volumes by the continuous agitation of the AgNP colloids and the surface-modified silica beads. The spectral scan of the individual AgNP-coated beads was performed to ensure the complete removal of the excess reducing agent trisodium citrate.

**Instrumental set-up**. The experimental platform integrates dual-trap optical tweezers manipulations and Raman spectroscopic measurements into an inverted microscope with the detailed instrumental layout shown in Supplementary Information (Supplementary Fig. 1). The dual-trap optical tweezers, force detection module, and bright field imaging are built-in the optical tweezers microscopy system (m-trap, LUMICKS, Netherlands). The 1064 nm laser source is splitted into two via a polarizing beamsplitter and focused inside the sample cell by a ×60 water immersion objective with a 1.2 numerical aperture (N.A.) for dual trapping, which are collected in the forward direction by position sensitive detector (PSD) for force analysis. A 780 nm light-emitting diode (LED) and a complementary metal oxide semiconductor (CMOS) camera are integrated into the beam path stereoscopically by a pair of 830 nm long-pass dichroic mirrors for bright field illumination and imaging, respectively. In addition, the 532 nm Raman excitation source (MLL-III-532-50 mW, CNI, China) is reflected by a notch filter (NF533-17, Thorlab, United

States) before entering the flex port of the optical tweezers microscope into its stereo double-layer-pathways with a 750 nm long-pass dichroic mirror for the recombination with the original two trapping laser beams. Adjusted by ND filters in the beam path to generate 25 mW input to the microscope, the power density at the Raman excitation focus is estimated to be $3.2 \times 10^6$ W/cm$^2$. The backscattered light is reflected by the 750 nm long-pass dichroic mirror and transmitted through the notch filter to enter a spectrometer (IsoPlane SCT-320, 1200 lines/mm, Teledyn Princeton Instrument, United States) with a liquid nitrogen-cooled charge-coupled device (CCD) camera (400B eXcelon, Teledyn Princeton Instrument, United States) at a spectral resolution of 2 cm$^{-1}$ for spectroscopic measurements. Besides, the flow cell at the sample stage is connected to a microfluidic system driven by compressed air to combine several microfluidic channels into several adjacent laminar fluidic streams at the center chamber during the experimental operations.

**Manipulation of two AgNP-coated beads and the in situ SERS measurement**. The experiments were conducted inside the microfluidic flow chamber with three adjacent laminar fluidic streams from AgNP-coated beads channel, buffer channel, and sample channel, shown in Supplementary Information (Supplementary Fig. 9). Moving the microfluidic flow cell by the equipped two-axis motorized translation micro-stage, different parts in the three adjacent laminar fluidic streams can be set at the place of the trapping laser beams and Raman excitation beam quickly. In the first step, the micro-stage moved the flow cell to a position that the focus of the trapping laser beams fall into the AgNP-coated beads stream to trap AgNP-coated beads. In the next step, the two AgNP-coated beads were trapped at the trapping laser beams and placed in the sample stream by the movement of the flow cell on the micro-stage. Prior to SERS measurement, spectra of solution in the sample stream were recorded as spectroscopic background. Spectra of individual trapped AgNP-coated beads under the Raman excitation spot were scanned to ensure neither intra-beads hotspots nor analyte attachments on the surface of AgNP-coated beads. Then, the two trapped AgNP-coated beads were approached in close proximity and the interparticle gap was placed at the spot for Raman excitation to correlate the relative bead distance to the spectroscopic detection. The motorized trapping laser positions, the camera-based bead distance estimations, and the trapping force detections were utilized to ensure the precise control of the two trapped AgNP-coated beads to create and adjust the SERS active structure under the microscopic visualization. When the two trapped AgNP-coated beads were set at constant separated distance and placed at the Raman excitation spot in the sample steam, the SERS spectra were recorded. All presented spectra were obtained upon the subtraction of the background accordingly and smoothed by Savitzky–Golay filter. Protein samples were dissolved and prepared in 1× PBS buffer, pH 7.4 for SERS measurements. The pH was measured before and after the SERS characterizations to ensure that the protein solutions maintained at physiological pH in the experiments. Ethanol was only used as the analyte in the first experiment whereas not involved in other experiments. The microfluidic flow chamber was cleaned thoroughly between experiments.

**Reporting summary**. Further information on experimental design is available in the Nature Research Reporting Summary linked to this paper.

## Data availability
The fabrication protocol of AgNP-coated beads is shared at Protocol Exchange[69]. Source data are provided[70], underlying Figs. 2, 3, 4, 5 and Supplementary Figures. The raw data that support the findings of this study are available in DataSpace at: https://doi.org/10.14711/dataset/DXO9VR. Source data are provided with this paper.

## Code availability
The spectral analysis and the simulation code of this study are available online at: https://doi.org/10.14711/dataset/DXO9VR.

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

## Acknowledgements
We acknowledge the funding support from Research Grant Council of Hong Kong under Project 26303018 and 16309919. We also acknowledge Prof. Chi-Ming Che and his funding support from "Laboratory for Synthetic Chemistry and Chemical Biology" under the Health@InnoHK Program launched by Innovation and Technology Commission, The Government of Hong Kong Special Administrative Region of the People's Republic of China.

## Author contributions
D.X., F.W., L.W., and H.J. constructed the experimental set-up and designed the experiments. D.X., F.W., C.H., M.V., Z.H., and L.C.W. performed the experiments and conducted the data analysis. D.X., F.W., and M.V. optimized the controllable SERS probe for spectroscopic measurements. C.H. fabricated and analyzed AgNP-coated beads. D.X., F.W., and Z.H. prepared alpha-synuclein. M.V., Z.H., and L.C.W. carried out the data acquisition and spectral analysis of ethanol, hemoglobin, and alpha-synuclein. L.W. and H.J. supervised the project. All authors participated in the result discussion and the manuscript preparation.

## Competing interests
The authors declare no competing interests.
