## [Peer Review File · Nature Communications]

Reviewers' Comments:

Reviewer #1:

Remarks to the Author:

Summary. The authors in the present manuscript have developed new optical tweezers-controlled Raman instrumentation suitable to investigate proteins using AgNPs probe and to test it on globular haemoglobin and amyloidogenic alpha-synuclein proteins.

General considerations: The manuscript is as SERS spectroscopy potentially interesting, but data interpretation is not accurate and sometimes wrong, so the manuscript needs an extensive revision.

Major concerns:

Title: The title should be changed because the term heterogeneity does not have a sense in protein structures. The power of SERS spectroscopy is to determine secondary protein structures in aqueous solution with high accuracy and at low concentration, respect other experimental techniques such as CD and NMR. Heterogeneity should be erased in the text.

Abstract.

The authors wrote, "Surface-enhanced Raman spectroscopy (SERS) probes molecular structures reaching the single-molecule level and empowering the label-free characterization of aqueous biomolecules." The problem is not label-free or probe. The concept is that both change the physical-chemistry of the studied molecule. The authors do not report data on this issue. Please change this issue in the text. Was demonstrated that NPs influence the aggregation kinetic of Abeta amyloid (doi: 10.1002/anie.201007824.)

Introduction:

Authors wrote: "...mainly in their amyloid aggregates stage...". For sake of clarity the authors should be describing IDPs aggregation process (doi: 10.1007/s00249-020-01424-1; 10.1016/j.bbamem.2018.02.022), SERS is especially useful to characterize oligomers, transient species before fibril formation. The authors must explain to the reader the kinetic process that leads proteins from monomers to fibrils; otherwise, confusion arises.

Authors wrote: "...achieved mainly focused on the fabrication of uniform nanostructures..." Change uniform nanostructures with more elegant "sharp size distribution".

Authors wrote: "Nevertheless, these procedures might alter the protein conformations during the complicated sample preparation and detection processes and are not ideal for characterizing the biomolecules in the physiological environment." This phrase has no sense because in literature is simple to find SERS measurement on cell.

Authors wrote: "The addition of biocompatible coatings on metal surface could alleviate this concern," This is not correct, because coating NPs decrease SERS effect since it vanishes after 2 nm.

Authors wrote: "Furthermore, SERS analysis of heterogenous alpha-synuclein at the physiological concentration was reported for the first time." The physiological concentration should be reported here, and 10 microM reported by the authors is wrong, its correct value is 1 microM (10.3389/fnins.2019.01399).

Results

Although laser ablation NPs (see ref 9) preparation is the best method to prepare NPs to investigate proteins secondary structures, the authors report AgNPs fabrication chemically using citrate to stabilize NPs. What is reducing agent used to prepare NPs, and how do they check the reducing agent has been removed especially in the case of haemoglobin? This issue has to be reported clearly in the main text because it is crucial for the investigations.

Fig. 4. CD spectra are too small, and it is impossible to see the legend. Authors wrote:few mixtures of unordered and helix, and the rare co-existence of helix and beta-sheet. This is wrong, because the co-existence is normally observed in IDPs (doi: 10.1002/smll.201500562; 10.1016/j.bbamem.2018.02.022).

Other points: The authors test their SERS measurements using CD spectra. CD spectra secondary structures determination is based on a database and have an intrinsic error over secondary structures determination of about 15-20 %. The best test is to compare the SERS intensity ratio between two specific peaks of the same spectrum (for example, intensity peak assigned to α -helix

and intensity peak assigned to β -sheet). The ratio of spectra reflects the ratio α -helix/ β -sheet of the considered globular model protein taken from the PDB database. In this way have an accurate validation of the used experimental technique.

The authors in the discussion section wrote: "More importantly, our method opens a new door to characterizing a small amount of biomolecules under diluted physiological conditions, particularly feasible for the heterogeneous systems.". Again, physiological conditions should be avoiding. Instead, the authors should be stress the fact that new optical tweezers-controlled Raman instrumentation has a great potential to investigate oligomeric state of amyloidogenic proteins since oligomers are transient species

Reviewer #2:

Remarks to the Author:

The article by Dai et al. reports on new configuration for SERS detection of biomolecules employing a multiple beams optical tweezers setup, two of which (1064nm) are used to trap and manipulate silica beads in order to form reproducible and controlled SERS hot spots, the third (532nm) to excite SERS scattering of the molecules in the hot spots. Applications are shown on Hgb and alpha synuclein. In this latter case the authors highlight the co-existence of several metastable structures of the protein.

The technique is nice and tailors to Ag-decorated silica beads, the original concept proposed by Svedberg on individual metal nanoparticles of creating hot spots by OT. The article is well written and timely.

This methodology could have far reaching applications in biosensing and be of interesting to the related scientific communities.

However, I feel that in the present state the data do not support fully the claims and I regret to say that I cannot recommend publication in the present form. Attached is a list of my concerns:

Major concerns

- Trapped particles are subject to brownian motion (both translational and rotational) that will displace the particles from one another in an uncontrolled way. This will change the gap among the nanoparticles forming the hot spots. This issue should be take into account. A position tracking of the beads particles should be provided to justify the stability of the gap size. Some calculations of the Brownian fluctuations could help to elucidate this issue that, in my opinion is key to assess reproducibility, since the SERS efficiency changes a lot even for nm variations of the gap size.
- The detection mechanism should be better elucidated. The particles are first approached and then the molecular flow is activated. The SERS is attributed to gap hot spots from Ag NPs located in the two silica beads. Since the signal depends on the exact location of the Raman laser, have the authors scanned the entire surface of a single nanoparticle (once exposed to the target molecule) to exclude that SERS comes from single particles ? Some plot of the SERS signal Vs time could help elucidate the origin of the signal. On both single and approached silica beads. What happens if, after observing SERS from a dimer, one of the particles is pulled away ?
- Some more detailed approach curves should be carried out in a repeated way to assess reproducibility (step size 10 nm or less).
- Figure 2a: The enhanced spectrum of EtOH (very nice this experiment!) should be plotted in a more extended range and compared to the Raman of EtOH. I do not agree with the attribution of the 1609 cm^{-1} feature to water. The Raman fingerprint of water in this region is a band not a sharp peak. Please investigate better.

- Force measured on the microbeads (figure 3b, page 9). It is not clear what the authors measure and how. Please clarify this important aspect.

- Raman spectrum and SERS spectra of Hgb:

-- The high intensity spectrum in Fig. 3a looks like amorphous carbon, indicating some molecular damage. Same holds for the spectra in Fig 5S.

-- The spectra shown in Fig. 3c look quite different from those in d and in Fig 4S. Why ? Unless this issue is afforded, it is difficult to make any assessment on the state of the molecule.

-- The Raman spectrum of Hgb does not typically show the broad bands in the 1300 and 1600 region (see e.g. Wood et al. DOI: 10.1016/B978-0-12-818610-7.00013-X or Casella et al DOI 10.1016/j.saa.2011.03.048). The origin of this intense signals should be assessed. Measurements as a function of power could help.

-- A very interesting range is from 200 to 800 cm^{-1} (Mizutani, DOI: 10.1016/B978-0-12-818610-7.00016-5). Can the authors show what happens also here ?

- Hgb reproducibility: the spectra are characterized by very sharp peaks superimposed to broad bands. I think that some careful analysis (histograms) should be carried out also on the smaller peaks since, in my opinion, any structural information is encoded also there.

- Sensitivity. Detecting Hgb at 10 μM is a good result but it is not what is expected by SERS in 2020. Reports in the literature show that 100nM and even 1 pM are concentration ranges are achievable (in this latter case by using an optical tweezers technique) and also single molecule SERS is feasible. The authors should explore the sensitivity limits of their technique prior to publication in Nat Commun.

- Physiological pH: this is always claimed but never measured. How does the presence of ethanol influence the structure of the molecules ?

- alfa syneuclein:

-- Figure 4b: Is the Raman spectrum of 2mM background subtracted ? Please comment.

-- I understand that 250 μM is below the sensitivity threshold of Raman, but some small features are visible, together with the Raman band of water (asterisk). Please comment. Also pls compare the Raman

-- SERS sensitivity. The authors show nice spectra at 10 μM . Some comparison with the SERS detection limits in the literature should be carried out and some exploration of the real limits of this technique on alfa-syneuclein should be a very valuable information.

-- Structure of the protein. I regret to say that this article does not provide any useful information on this issue. Indeed the spectra vary, and this can be a sign of conformational changes, but this is obvious and does not add valuable information. The authors are not able to drive the experimental conditions so to control and probe the structure in each case.

-- Some comparison between the a 10 μM Raman and the SERS should be given.

Minor concerns

- Please change "at the physiological concentration" with "at physiological concentration" whenever it applies.

Page 3, second paragraph. "This dynamic SERS detection has great tunability in the microfluidic flow ..." Please specify better this point. What is versatile in the microfluidic cell ?

Reviewer #3:

Remarks to the Author:

X. Dai and co-authors report on a new SERS probe for protein spectroscopy. The probe is based on optical tweezers where two laser beams are used to control the gap distance between two Ag decorated nanoparticles.

The idea to use the double laser beam to control the trapping and the SERS enhancement is nice and, accordingly to the results presented in the manuscript, it seems to work.

Anyway, in my opinion the manuscript in the present form is not suitable for nature communications (maybe it can be for a sister mid-impact journal).

First of all the authors discuss the potential application of SERS in single molecule spectroscopy. This is a well known research field and recently some important results have been published also in Nat. Comm (see for instance Nat. Commun. 9, 1733 (2018) and NATURE COMMUNICATIONS (2019) 10:5321). In particular the demonstration of single molecule protein spectroscopy in SERS platform integrated with optical / plasmonic tweezers has been reported in doi.org/10.1002/anie.202000489.

The results presented in the submitted manuscript is rather far from these state-of-the-art methods. In fact no single molecule sensitivity is discussed. Moreover it's not clear if the platform can reach this sensitivity. Considering the calculated EF it seems not the case.

The manuscript then illustrates three different experiments on SERS with the proposed optical tweezers. The first one is a very simple Raman spectroscopy of EtOH and it clearly demonstrates that they can control the distance between two beads in order to achieve high EF. Anyway, the authors claim some sub-nm control of the distance but no demonstration of that is reported. This is a very crucial point.

In a second experiment the authors use the same platform to perform SERS on hemoglobin. Figure 3 illustrates some reproducibility tests and reports (fig. 3b) some number for trapping force. Anyway, also here no details are reported. Looking at the experimental spectra it seems that they can just detect few main Raman peaks (with long acquisition time (5 sec) and no so low concentration (5 microM)).

In the final experiment the authors want to demonstrate that the proposed platform can be used to investigate on the conformational states of alpha-synuclein at physiological concentration. To me this part is also very critical and MUST be discussed taking into consideration the following points:

The authors claim that it's possible to discriminate alfa-helical, beta-sheet and random-coil conformations. The protein itself, presents sections that are mainly alfa-helical and beta-sheet, as well as random coil. To me, in a SERS experiment where nanogaps between Ag-nanoparticles are used as Hot-Spots, it's not possible to know if the difference in the spectra are due to different section of the protein interacting with the hot-spot or are due to different protein conformations. The size of the hot-spot, in fact, can be very small and comparable to sub-section of the protein, so justifying different spectrum for different measurements. This phenomena is very well discussed in doi.org/10.1002/anie.202000489. In order to demonstrate that the platform is able to detect protein conformation, the authors should consider additional experiments where different proteins (at least 2 with well known conformations) are used.

In conclusion, the idea of 2-laser optical tweezers and controlled SERS is very good and promising, but the manuscript, in order to be published in a high impact journal must be better presented (also the quality of the figures are really bad!). A deeper discussion on the tweezing system to "sub-nm" control the gap and additional experiments on sensitivity are needed.

We thank the reviewers for their insightful critiques, and we have made the following changes in response to their helpful comments and suggestions:

Reviewer #1 (Remarks to the Author):

Summary. The authors in the present manuscript have developed new optical tweezers-controlled Raman instrumentation suitable to investigate proteins using AgNPs probe and to test it on globular haemoglobin and amyloidogenic alpha-synuclein proteins.

General considerations: The manuscript is as SERS spectroscopy potentially interesting, but data interpretation is not accurate and sometimes wrong, so the manuscript needs an extensive revision.

Major concerns:

1. Title: The title should be changed because the term heterogeneity does not have a sense in protein structures. The power of SERS spectroscopy is to determine secondary protein structures in aqueous solution with high accuracy and at low concentration, respect other experimental techniques such as CD and NMR. Heterogeneity should be erased in the text.

We have changed the title to “*Optical tweezers-controlled hotspot for sensitive and reproducible SERS characterization of native protein structures*” from “Optical tweezers-controlled SERS probe identifies the heterogeneity of protein conformations at physiological concentration”. We would like to showcase the advantages of controlling the SERS active interparticle gap between the two AgNP-coated beads by the optical tweezers-coupled Raman spectroscopy and highlight the feasibility of this SERS platform in analyzing the protein structures at native states, especially for the transient species of intrinsically disordered proteins, in dilute aqueous solutions. Referring to this comment, we have revised the entire manuscript with new experimental results to demonstrate the performance of our SERS platform for protein structure characterizations, in term of sensitivity, accuracy, and reproducibility.

2. Abstract.

The authors wrote, “Surface-enhanced Raman spectroscopy (SERS) probes molecular structures reaching the single-molecule level and empowering the label-free characterization of aqueous biomolecules.” The problem is not label-free or probe. The concept is that both change the physical-chemistry of the studied molecule. The authors do not report data on this issue. Please change this issue in the text. Was demonstrated that NPs influence the aggregation kinetic of Abeta amyloid (doi: 10.1002/anie.201007824.)

We have rewritten the sentence to “*Surface-enhanced Raman spectroscopy (SERS) has emerged as a powerful tool to detect biomolecules in aqueous environments.*” If the nanoparticles and biomolecules co-aggregated for a long interaction time, the physical or chemical properties of the biomolecule could be affected. However, we used mechanical control instead of chemical capping agents to create hotspot then place the SERS active interparticle gap between the two trapped AgNP-coated beads in the microfluidic flow chamber as a detection window to detect the freely passing-by proteins. Perturbations to the protein can be minimized in our platform by controlling the flowing rate and the Raman excitation power. The two trapped AgNP-coated beads could be replaced freshly in parallel SERS measurements, to further minimize the interaction between metal nanoparticles and proteins. Our results show that the oxidation state, coordination number, and spin state of hemes in hemoglobin were well-preserved in the flash spectroscopic detections. Moreover, we have conducted additional SERS characterizations on two extra typical globular proteins: lysozyme and bovine serum albumin (BSA), showing their native conformations consistent with the previous study. In particular, a study shows that metal nanoparticles without immobilizing agents or aggregating salts did not affect neither the tertiary nor the secondary structures of BSA (doi: 10.3389/fchem.2019.00030) and confirming that the short interaction time between metal nanoparticles and protein solution did not affect the protein secondary structures. (doi: 10.1039/C7CP08552D)

Moreover, different nanoparticles might influence the aggregation kinetics of different amyloid proteins. The suggested reference reports that the inorganic nanoparticles CdTe prevented the fibrillation of amyloid peptides A β , due to the multiple bindings between them. (doi: 10.1002/anie.201007824.) Whereas, another reference suggests that Ag nanoparticles just slightly influenced the fibril formation kinetics of A β and hIAPP. (doi: 10.1039/C7CP08552D) Analogous to the latter, we expect that our AgNP-coated beads would not significantly affect the aggregation kinetics of alpha-synuclein because the interaction between metal nanoparticles and proteins was minimized in our optimized experimental setting. More importantly, the focus of the manuscript is the structural characterization of different species of alpha-synuclein co-existing at a single time-point, not involving the kinetic analysis for the growth of amyloid fibrils. We might conduct kinetic analysis of the amyloid fibril formation of alpha-synuclein at more time-points and consider this potential influence in the futur.

3. Introduction:

Authors wrote: “.....mainly in their amyloid aggregates stage...”. For sake of clarity the authors should be describing IDPs aggregation process (doi: 10.1007/s00249-020-01424-1; 10.1016/j.bbamem.2018.02.022), SERS is especially useful to characterize oligomers, transient

species before fibril formation. The authors must explain to the reader the kinetic process that leads proteins from monomers to fibrils; otherwise, confusion arises.

We highly appreciate this helpful suggestion. We have added the description of IDPs aggregation process and emphasized the feasibility of SERS for the characterization of the significant IDP oligomers, based on the previous SERS studies on analogous amyloid proteins: A β and hIAPP (doi: 10.1007/s00249-020-01424-1; 10.1016/j.bbamem.2018.02.022) as well as the literature reviews of alpha-synuclein (doi: 10.1038/nrn3406).

“Inheriting these advantages, surface-enhanced Raman spectroscopy (SERS) boosts up the sensitivity to detect proteins at low concentrations, even at the single-molecule level, to mimic physiological conditions^{4, 5, 6}. It is also a powerful tool to characterize the dynamic ensembles of variable conformations of intrinsically disordered proteins (IDPs)^{7, 8, 9, 10}. IDPs lack stable secondary and tertiary structures as monomers in aqueous environments, but sometimes self-assemble into oligomers with various structures and further grow into amyloid fibrils¹¹, which are associated with the incurable neurodegenerative diseases¹². The dynamic conversion from monomers to oligomers is the key step in the early pathological development¹³. However, the transient nature and the low population of oligomers make it challenging to characterize their structural features, which are responsible for the cellular toxicity and the on-going amyloid aggregation¹⁴. In particular, the SERS study on alpha-synuclein, an IDP closely related to Parkinson’s disease^{15, 16, 17}, is scarce¹⁸.”

4. Authors wrote: “...achieved mainly focused on the fabrication of uniform nanostructures....” Change uniform nanostructures with more elegant “sharp size distribution”.

We have rewritten the sentence to *“Advancements have been achieved mainly on fabricating SERS substrates with sharp size distribution^{27, 28, 29}, in order to provide consistent SERS enhancements.”*

5. Authors wrote: “Nevertheless, these procedures might alter the protein conformations during the complicated sample preparation and detection processes and are not ideal for characterizing the biomolecules in the physiological environment.” This phrase has no sense because in literature is simple to find SERS measurement on cell.

Authors wrote: “The addition of biocompatible coatings on metal surface could alleviate this concern,” This is not correct, because coating NPs decrease SERS effect since it vanishes after 2 nm.

We have deleted the above two inaccurate statements, rephrased the limitations of using the conventional salt-induced nanoparticle-protein aggregates, and underlined the feasibility of SERS in bio-analysis.

Revised main text on the limitations of using the conventional salt-induced nanoparticle-protein aggregates:

“Besides, practical difficulties emerge when using conventional nanoparticle-based SERS substrates generated from salt-induced random aggregations to probe proteins in dilute solutions, such as low efficiency, poor reproducibility, and potential structural perturbation during long incubation time in SERS measurements^{19, 20}.”

“Nevertheless, when measuring analytes at low concentrations, it is inconvenient to locate and focus on the nanometer-size hotspots from the conventional aggregated nanoparticles in colloidal suspension for in situ detection, thus lowering the feasibility and efficiency of the technique.”

Revised main text on the feasibility of SERS in bio-analysis:

“Surface-enhanced Raman spectroscopy (SERS) has emerged as a powerful tool to detect biomolecules in aqueous environments.”

“Inheriting these advantages, surface-enhanced Raman spectroscopy (SERS) boosts up the sensitivity to detect proteins at low concentrations, even at the single-molecule level, to mimic physiological conditions^{4, 5, 6}. It is also a powerful tool to characterize the dynamic ensembles of variable conformations of intrinsically disordered proteins (IDPs)^{7, 8, 9, 10}.”

“Smith and co-workers trapped one partially Ag-coated silica microparticle as a mobile and visible SERS probe⁴⁵, which is further developed by Petrov and co-workers to accomplish interfacial detection of living cells⁴⁶.”

6. Authors wrote: "Furthermore, SERS analysis of heterogenous alpha-synuclein at the physiological concentration was reported for the first time." The physiological concentration should be reported here, and 10 microM reported by the authors is wrong, its correct value is 1 microM (10.3389/fnins.2019.01399).

We thank the reviewer pointing out a more recent and accurate physiological concentration of alpha-synuclein as 1 μ M. We have revised the relating content and re-performed the SERS measurements of alpha-synuclein at 1 μ M to replace the old results at 10 μ M. We are able to obtain high quality SERS spectra of alpha-synuclein in 1 μ M aqueous solution with low laser

power and short accumulation time, owing to the high sensitivity and reproducibility of our SERS platform. All the related figures and contents are revised accordingly.

Revised main text on the physiological concentration of alpha-synuclein:

“In particular, the SERS characterization of alpha-synuclein at physiological concentration ($1 \mu\text{M}$)⁴⁷ was reported for the first time, showing the structural variations arisen from its transient species.”

“The physiological concentration of alpha-synuclein at non-aggregated states is $1 \mu\text{M}$ ⁴⁷, below the detection threshold of Raman spectroscopy. Whereas the experimental limit of detection (LOD) of alpha-synuclein on our SERS platform is 100 nM, thus this sensitive SERS approach is feasible to characterize the transient species of alpha-synuclein in dilute solutions.”

Revised Fig. 5 showing the new SERS spectra of $1 \mu\text{M}$ alpha-synuclein aqueous solution:

Figure 5. The spectroscopic characterizations of an intrinsically disordered protein: alpha-synuclein. **a** CD spectrum of 200 μM alpha-synuclein in aqueous solution. **b** Spontaneous Raman spectra of 2 mM (green) and 250 μM (blue) alpha-synuclein solution with 10 min acquisition time. **c** The comparison among three representative types of SERS spectra of 1 μM alpha-synuclein solution with 1 s acquisition time (blue, red, and black) and the SERS spectrum of 250 μM alpha-synuclein solution with 5 min acquisition time (purple). **d** Mapping of 200 SERS spectra of 1 μM alpha-synuclein solution obtained from two AgNP-coated beads trapped at 20 nm with 1 s acquisition time. The color bar shows the normalized intensities from low (dark blue) to high (red). **e** Illustration of the ensemble averaging from the measurement of alpha-synuclein at high concentration with long accumulation time and the small-size

sampling from the measurements of alpha-synuclein at low concentration with short accumulation time.”

7. Results

Although laser ablation NPs (see ref 9) preparation is the best method to prepare NPs to investigate proteins secondary structures, the authors report AgNPs fabrication chemically using citrate to stabilize NPs. What is reducing agent used to prepare NPs, and how do they check the reducing agent has been removed especially in the case of haemoglobin? This issue has to be reported clearly in the main text because it is crucial for the investigations.

We agree that laser ablation is better than chemical fabrication to prepare Ag nanoparticles, but the latter is more accessible and convenient. One of our improvements comes from avoiding chemical capping agents to induce the random aggregations of Ag nanoparticles and solely utilizing mechanical manipulation to create SERS active hotspots as an alternative. In our study, we used trisodium citrate as the reducing agent to prepare AgNPs. Trisodium citrate is commonly used in the fabrication of AgNPs, which can act as both a reducing and a stabilizing agent. Before coating the AgNPs to the silica beads, the AgNPs colloid was washed with Milli-Q ultrapure water for three times, in order to remove the excess reducing agent. In the SERS measurement, we trapped the AgNP beads in the microfluidic buffer flow, which would further wash the reducing agent away. To verify it, we have acquired the SERS spectra of the individual AgNP beads in Fig. 10S, where the vibrational signal of citrate was not observed. The vibrational signal of citrate was not observed in the SERS spectra of hemoglobin, either. The spectral features of hemoglobin indicate its native states without any influence, in terms of the oxidation state, the coordination number, and the spin state of hemes. Thus, the reducing agent trisodium citrate has been removed completely before the SERS characterization of proteins.

New description added in the main content:

“The AgNPs were prepared from the reduction of silver nitrate by trisodium citrate and washed three times to remove the reducing agent before mixing with the modified silica beads.”

More detailed description added in the Material and Method section:

“Fabrication of AgNP-coated beads

AgNP-coated beads were prepared by coating the AgNPs on the surface of silica beads ($R = 0.63 \mu\text{m}$)^{45, 46}. First, the silver nanoparticles were synthesized based on the conventional method with optimized parameters^{3, 50}. 50 mL 1 mM AgNO_3 aqueous solution was heated to boiling, followed by the drop by drop addition of 1.0 mL 0.1 M trisodium citrate solution. The mixture was kept boiling for 16 min under constant stirring, then was cooled down to room

temperature showing yellowish gray color. The AgNPs colloid was washed with Milli-Q ultrapure water for three times, in order to remove the excess reducing agent. Meanwhile, the surface of the micrometer-size silica beads was modified for the AgNP coating. Silica beads stock (5.0% w/v) was dried at 60 °C before dispersed in anhydrous ethanol. Then, a 0.2% ethanol solution of (3-aminopropyl)triethoxysilane (APTES) was added into beads suspension (final concentration 0.1%) for 24 h shaking incubation at room temperature. The solution was purified by centrifugation with distilled ethanol at 3500 rpm for three times to discard the supernatant. After drying at 60 °C to remove the ethanol and adding double-distilled water, the silica beads with the amino groups modified surface were obtained. Lastly, the AgNP-coated beads were prepared in 1 mL volumes by the continuous agitation of the AgNP colloids and the surface-modified silica beads. The spectral scan of the individual AgNP-coated beads was performed to ensure the completely removal of the excess reducing agent trisodium citrate.”

New experimental results added in Fig. 10S:

Figure 10S. SERS spectra of individual trapped AgNP-coated beads at the Raman excitation spot in the sample stream of 100 nM hemoglobin. The blank spectra confirmed neither intra-bead hotspots were formed nor hemoglobin molecules were attached on the surface of beads. The reducing agent trisodium citrate was completely removed to avoid interference.”

8. Fig. 4. CD spectra are too small, and it is impossible to see the legend. Authors wrote:few mixtures of unordered and helix, and the rare co-existence of helix and beta-sheet. This is wrong, because the co-existence is normally observed in IDPs (doi: 10.1002/smll.201500562; 10.1016/j.bbamem.2018.02.022).

We have re-drawn the CD spectra and revised the inaccurate statement. To better illustrate the co-existence of different IDP structures, we have conducted new SERS experiments on two typical globular proteins (lysozyme and bovine serum albumin) that exist in the homogenous structures from one to another for comparison.

The suggested reference summarizes the significantly variable conformations of A β and Islet Amyloid Polypeptide (IAPP) in the amyloid formation and membrane disruption process. (DOI: 10.1016/j.bbamem.2018.02.022) It is true that the co-existence of different species with various structures is normally observed in the bulk measurements of IDPs, which could be further deconvoluted to resolve the structural distribution. We are able to obtain similar results from the spontaneous Raman measurement of 2 mM alpha-synuclein solution with 10 min acquisition time and the SERS measurement of 250 μ M alpha-synuclein solution with 5 min acquisition time. Figures and spectral analysis are shown in the next response to quantify the co-existence of diverse conformations of alpha-synuclein.

Furthermore, the suggested reference demonstrates the heterogenous structures of individual hIAPP amyloid fibrils characterize by Tip-Enhanced Raman Spectroscopy with high spatial resolution, which has already been cited in the manuscript. (doi: 10.1002/sml.201500562) Analogous to this mechanical control of hotspot at the designated position, our platform could manipulate two AgNP-coated beads in the microfluidic protein flow to create the dynamic hotspot as a sensitive SERS detection window to characterize the passing-by proteins. Thus, we could directly identify the structural variations arisen from different species of alpha-synuclein in the parallel SERS measurements using the dilute solution (1 μ M) and short accumulation time (1 s). Among 200 parallel small-size sampling SERS measurements, the amide I and the amide III bands of the SERS spectra of 1 μ M alpha-synuclein exhibit prominent variations in comparison to the uniform spectral features of 1 μ M lysozyme, indicating the co-existence of the monomeric species and the transient species of alpha-synuclein in aqueous solution at physiological concentration.

Newly added Fig.4, updated Fig. 5, and the revised main text to demonstrate the co-existence of different alpha-synuclein species with various structures:

Figure 4. The spectroscopic characterizations of lysozyme in its compact globular structure. **a** The comparison between 50 SERS spectra of 1 μ M lysozyme solution with 1 s acquisition time

(bottom) and the spontaneous Raman spectrum of 1 mM lysozyme solution with 5 min acquisition time (top). **b** Histogram of the Amide I band distribution of the 50 SERS spectra of 1 μM lysozyme solution in **a**, indicating the mean as 1655 cm^{-1} with 0.1% RSD. **c** Illustration of the ensemble averaging from the spontaneous Raman measurement of lysozyme in the concentrated solution and the small-size sampling from the SERS measurements of lysozyme in the dilute solution.”

Figure 5. The spectroscopic characterizations of an intrinsically disordered protein: alpha-synuclein. **a** CD spectrum of 200 μM alpha-synuclein in aqueous solution. **b** Spontaneous Raman spectra of 2 mM (green) and 250 μM (blue) alpha-synuclein solution with 10 min acquisition time. **c** The comparison among three representative types of SERS spectra of 1 μM alpha-synuclein solution with 1 s acquisition time (blue, red, and black) and the SERS spectrum of 250 μM alpha-synuclein solution with 5 min acquisition time (purple). **d** Mapping of 200 SERS spectra of 1 μM alpha-synuclein solution obtained from two AgNP-coated beads trapped at 20 nm with 1 s acquisition time. The color bar shows the normalized intensities from low (dark blue) to high (red). **e** Illustration of the ensemble averaging from the measurement of alpha-synuclein at high concentration with long accumulation time and the small-size sampling from the measurements of alpha-synuclein at low concentration with short accumulation time.”

“Fig. 5c demonstrates three representative types of SERS spectra of 1 μM alpha-synuclein solution with 1 s acquisition time among the 200 parallel experiments shown in Fig. 5d. Strikingly, the amide I and the amide III bands of the SERS spectra of 1 μM alpha-synuclein exhibit prominent variations in comparison to the uniform spectral patterns of 1 μM lysozyme

in Fig. 4, indicating the co-existence of different alpha-synuclein species with various structures at physiological concentration⁷.”

9. Other points: The authors test their SERS measurements using CD spectra. CD spectra secondary structures determination is based on a database and have an intrinsic error over secondary structures determination of about 15-20 %. The best test is to compare the SERS intensity ratio between two specific peaks of the same spectrum (for example, intensity peak assigned to α -helix and intensity peak assigned to β -sheet). The ratio of spectra reflects the ratio α -helix/ β -sheet of the considered globular model protein taken from the PDB database. In this way have an accurate validation of the used experimental technique.

We thank the reviewer for suggesting a good validation method. we have conducted new complementary SERS analysis on two typical globular proteins (lysozyme and bovine serum albumin) to compare the secondary structure components deconvoluted from our SERS spectra to the literature data, in order to validate the accuracy and reliability of the protein structural characterization on our SERS platform.

Specifically, the spectral deconvolution of two typical globular proteins (lysozyme and bovine serum albumin) was conducted and plotted in Fig 12S and 14S. These globular proteins are known to contain different structural sections in individual protein and possess the identical folded structure from one to another with rich structural information characterized by different techniques in the PDB database. The spectral deconvolution of the SERS spectrum of lysozyme (45.2% α -helix, 11.3% β -sheet and 43.5% random coil) is similar to the X-ray diffraction data (29–42% α -helix, 10% β -sheet, 48–62% random coil) (doi.org/10.1073/pnas.57.3.483) and consistent with other literature data: (32% α -helix, 9% β -sheet, 59% random coil from Raman spectra; 26–29% α -helix, 16-11% β -sheet, 55–68% random coil from CD spectra. [doi: 10.1021/ja00438a057](https://doi.org/10.1021/ja00438a057)). Meanwhile, the spectral deconvolution of bovine serum albumin (66.5% α -helix, 9.5% β -sheet and 24.1% random coil) also well matches literature data (68% α -helix, 17% β -sheet and 15% random coil from X-ray diffraction data; 53% α -helix, 14% β -sheet and 16% random coil from FTIR spectra; 54% α -helix, 18% β -sheet and not definite random coil from CD spectra. [doi:10.1088/1742-6596/769/1/012016](https://doi.org/10.1088/1742-6596/769/1/012016)). Hence, the SERS characterization on our platform can provide accuracy assessment on the secondary structure components of globular proteins.

New Fig. 12S and Fig. 14S demonstrating the analysis on the SERS spectra of two typical globular proteins (lysozyme and bovine serum albumin):

Figure 12S. Deconvolution of SERS signal of 1 μM lysozyme in amide I region. The composition of different secondary structure, such as α -helix (45.2%), β -sheet (11.3%) and random coil (43.5%), is consistent with the previous investigations^{4,5}.”

Figure 14S. Deconvolution of 1 μM BSA SERS signal in amide I region. The composition of different secondary structure, such as α -helix (66.5%), β -sheet (9.5%) and random coil (24.1%), is consistent with the previous studies⁶.”

Newly added description on the spectral analysis on two typical globular proteins (lysozyme and bovine serum albumin):

“The spectral contributions from different secondary structures were deconvolved to α -helix (45.2%), β -sheet (11.3%) and random coil (43.5%) in Fig. 12S, which is consistent with the previous investigations^{61, 62}.”

“Moreover, the SERS spectra of 1 μ M BSA in Fig. 13S and Fig. 14S also provide the structural component assessment (66.5% α -helix, 9.5% β -sheet and 24.1% random coil) supported by the previous studies⁶³.”

10. The authors in the discussion section wrote: “More importantly, our method opens a new door to characterizing a small amount of biomolecules under diluted physiological conditions, particularly feasible for the heterogeneous systems.”. Again, physiological conditions should be avoiding. Instead, the authors should stress the fact that new optical tweezers-controlled Raman instrumentation has a great potential to investigate oligomeric state of amyloidogenic proteins since oligomers are transient species

We highly appreciate this suggestion. We have revised the statement to emphasize the feasibility of SERS to characterize the significant but transient oligomers of amyloidogenic proteins in the amyloid formation process.

Revised discussion section on the potential applications of our SERS platform:

“More importantly, our method opens a new door to characterize the unique structural variations of IDPs in dilute solutions, which remains a significant challenge in the biophysics community. This dynamic SERS probe has great potential to investigate the oligomeric state of amyloidogenic proteins and resolve the structural details as well as the conformational population of these transient species prior to the amyloid aggregation, providing profound molecular insights to understand the onset of neurodegenerative diseases.”

Reviewer #2 (Remarks to the Author):

The article by Dai et al. reports on new configuration for SERS detection of biomolecules employing a multiple beams optical tweezers setup, two of which (1064nm) are used to trap and manipulate silica beads in order to form reproducible and controlled SERS hot spots, the third (532nm) to excite SERS scattering of the molecules in the hot spots. Applications are shown on Hgb and alpha synuclein. In this latter case the authors highlight the co-existence of several metastable structures of the protein.

The technique is nice and tailors to Ag-decorated silica beads, the original concept proposed by Svedberg on individual metal nanoparticles of creating hot spots by OT. The article is well written and timely.

This methodology could have far reaching applications in biosensing and be of interesting to the related scientific communities.

However, I feel that in the present state the data do not support fully the claims and I regret to say that I cannot recommend publication in the present form. Attached is a list of my concerns:

Major concerns

1. - Trapped particles are subject to brownian motion (both translational and rotational) that will displace the particles from one another in an uncontrolled way. This will change the gap among the nanoparticles forming the hot spots. This issue should be take into account. A position tracking of the beads particles should be provided to justify the stability of the gap size. Some calculations of the Brownian fluctuations could help to elucidate this issue that, in my opinion is key to assess reproducibility, since the SERS efficiency changes a lot even for nm variations of the gap size.

We agree with the reviewer that the distance stability is the key to assess the reproducibility. As suggested, we have conducted the position tracking of the two trapped beads and performed the computational simulation of the Brownian fluctuation of the trapped bead. The experimental result and the theoretical calculation are in excellent agreement. Since the shortest spectroscopic accumulation time is 1 second, the translational and rotational motions of the two trapped beads are averaged out to generate the reproducible SERS enhancements between the interparticle gaps over times. When $R=0.63 \mu\text{m}$ and $k=2.0 \text{ pN/nm}$, the average bead positions in 1 second over the time trace of 15 seconds at the sampling rate of $\Delta t=10^{-2} \text{ s}$ were simulated, giving the position fluctuation as 0.12 nm. In the experiment, the average bead distances in 1 second over the time trace of 15 seconds at the sampling rate of $\Delta t=3 \times 10^{-2} \text{ s}$ were recorded, maintaining the distance fluctuation within the SD of 0.29 nm.

The motion of an optically trapped Brownian particle in one dimension can be modeled by the Langevin equation (<https://doi.org/10.1119/1.4772632>):

$$\underbrace{m\ddot{x}(t)}_{\text{inertia}} = \underbrace{-\gamma\dot{x}(t)}_{\text{friction}} + \underbrace{kx(t)}_{\text{restoring force}} + \underbrace{\sqrt{2k_B T \gamma} W(t)}_{\text{white noise}}, \quad (1)$$

where x is the particle position, m is its mass, γ is the friction coefficient, k is the trap stiffness, $\sqrt{2k_B T \gamma} W(t)$ the fluctuating force due to random impulses from the many neighboring fluid

molecules, k_B is Boltzmann's constant, and T is the absolute temperature. This equation can be written as:

$$\dot{\vec{r}}(t) = -\frac{1}{\gamma} \vec{k} \cdot \vec{r}(t) + \sqrt{2D} \vec{W}(t), \quad (2)$$

where $\vec{r} = [x, y, z]$ represents the position of the particle. Inserting $\gamma = 6\pi\eta R$ and $D = \frac{k_B T}{\gamma}$

into the equation (2), we can simulate the position of one trapped bead at the trap stiffness $k=2.0$ pN/nm and the bead radius $R=0.63$ μm in 1 second at the sampling rate of $\Delta t=10^{-6}$ s, generating the standard deviation (SD) as 1.54 nm. Since the shortest spectroscopic accumulation time is 1 second, the average bead positions in 1 second over the time trace of 15 seconds were plotted in Fig.3S a, demonstrating the stable and reproducible placements of the trapped beads at the designated position (0 nm) with the fluctuation as 0.12 nm.

In the experiment, we trapped two beads at the distance of 20 nm and recorded the time trace of the bead distance over 15 seconds. The fluctuation of the bead distance in 15 seconds at the sampling rate of $\Delta t=3 \times 10^{-2}$ s was analyzed to give the SD of 1.65 nm. Since the shortest spectroscopic accumulation time is 1 second, the translational and rotational motions of the two trapped beads are averaged out to generate the reproducible SERS enhancements between the interparticle gaps over times. Fig. 3S b shows the average bead distances in 1 second over the time trace of 15 seconds at the sampling rate of $\Delta t=3 \times 10^{-2}$ s at 20 nm within the SD of 0.29 nm. Hence, the two beads can be trapped by optical tweezers at the precise positions over times to ensure the SERS reproducibility, which is supported by both the theoretical simulation and the experimental measurement.

Newly added Fig. 3S to demonstrate the bead trapping stability.

Figure 3S. a: The theoretical simulation of the Brownian fluctuation of one trapped bead ($R=0.63$ μm , $k=2.0$ pN/nm, sampling rate $\Delta t=10^{-2}$ s), giving the $SD=0.12$ nm for the average

bead positions in 1 second over 15 seconds. b: The experimental time traces of the distance between the two AgNP-coated beads trapped at 20 nm at the sampling rate of $\Delta t = 3 \times 10^{-2}$ s, demonstrating the $SD = 0.29$ nm for the average bead distance in 1 second over 15 seconds."

2. - The detection mechanism should be better elucidated. The particles are first approached and then the molecular flow is activated. The SERS is attributed to gap hot spots from Ag NPs located in the two silica beads. Since the signal depends on the exact location of the Raman laser, have the authors scanned the entire surface of a single nanoparticle (once exposed to the target molecule) to exclude that SERS comes from single particles ? Some plot of the SERS signal Vs time could help elucidate the origin of the signal. On both single and approached silica beads. What happens if, after observing SERS from a dimer, one of the particles is pulled away ?

We have conducted new additional measurements to scan the entire surface of AgNP-coated bead in the sample channel before and after the creation of the SERS active interparticle hotspot. These results are added to the supplementary information to elucidate that the origin of the SERS signal was from the controllable hotspot between the two approached AgNP-coated beads. We have also revised the description and the figures on the detection procedure accordingly.

Specifically, we trapped one AgNP-coated bead and moved it to the Raman excitation spot inside the 1% ethanol sample channel to measure the time-series SERS spectra, which were recorded directly upon the subtraction of the spontaneous Raman spectrum of 1% ethanol solution as background in Fig. 1R. Since the sizes of the AgNP-coated bead and that of the Raman excitation spot were comparable at around 1 μm and the trapped AgNP-coated bead was in rapid rotation, the entire bead surface was well characterized while it was trapped at the Raman excitation spot for continuous 20 seconds, generating 20 spectra without obvious vibrational signals of ethanol in Fig. 6S b. Then we repeated the measurement on the second AgNP-coated bead to obtain similar blank spectra, indicating no active intra-bead hotspots on the bead surface. When the two AgNP-coated beads were approached at 0 nm under the Raman excitation spot, the spectral signal of ethanol emerged, generating 20 SERS spectra in continuous 20 s in Fig. 6S c. After the hotspot creation, we moved one bead to the center of the Raman excitation spot and pulled the other one away for time-series spectroscopic measurements. 20 continuous spectra were recorded in Fig. 6S d, where the vibrational characteristics of ethanol were not observed. We repeated the measurement on the other AgNP-coated bead and obtained similar blank spectra. These experimental results verify that the significant SERS enhancement was generated between the two approached AgNP-coated beads, not from the surface of individual AgNP-coated beads.

The revised main text on the spectroscopic scans of individual AgNP-coated beads:

“To confirm the SERS enhancement effect from the hotspot between the two approached AgNP-coated beads, the entire surface of individual AgNP-coated beads was scanned in time-series measurements. The blank spectra of the single AgNP-coated bead trapped at the Raman excitation spot before and after the hotspot creation in Fig. 6S imply no intra-bead hotspots.”

Newly added Fig. 6S on the spectroscopic scans of individual AgNP-coated beads:

“

*Figure 6S. Time series SERS spectra of single and approached AgNP-coated beads in 1% ethanol solution. **a**: Schematic diagram of one AgNP-coated bead trapped under Raman excitation spot, where the sizes of the AgNP-coated bead and that of the Raman excitation spot were comparable at around 1 μm and the tapped AgNP-coated bead was in rapid rotation. **b**: Time series SERS spectra of one AgNP-coated bead trapped at the Raman excitation spot in 1% ethanol solution. **c**: Time series SERS spectra of two AgNP-coated beads approached at 0 nm under the Raman excitation spot in 1% ethanol solution. **d**: Time series SERS spectra of one AgNP-coated bead trapped at the Raman excitation spot in 1% ethanol solution when the other bead was moved away after the creation of SERS active interparticle hotspot in **c**. The time series SERS spectra in **b**, **c**, **d** were recorded in continuous 20 s with the acquisition time of 1 s per spectrum. The blank spectra in **b** and **d** indicated that no active intra-bead hotspot was observed. **e** and **f**: Histogram of peaks intensities of ethanol at 886 cm^{-1} (mean=34497.40, RSD=11.9%) and 1458 cm^{-1} (mean=19137.55, RSD=15.9%) for the 20 SERS spectra in **d**, indicating the stability of enhancement within the created hotspot < 16%.”*

The reference of the background for spectral subtraction:

Figure 1R. Typical Raman spectra of 1% ethanol solution with 1 s acquisition time as background, showing a broad water band at $\sim 1640\text{ cm}^{-1}$.

Revised Fig. 9S to illustrate the detection procedure:

Figure 9S. **a**: Illustration of the microfluidic flow cell with three adjacent laminar fluidic streams from sample channel, buffer channel, and AgNP-coated bead channel. **b**: Illustration

of the detection procedure: Step 1: Two AgNP-coated beads are trapped at AgNP-coated beads stream. Step 2: Two trapped AgNP-coated beads are placed at sample stream. Step 2.5: Individual trapped AgNP-coated beads is brought to the Raman spot for spectroscopic scan. Step 3: Two AgNP-coated beads are approached to generate SERS enhancement for sample measurements.”

3. - Some more detailed approach curves should be carried out in a repeated way to assess reproducibility (step size 10 nm or less).

- Figure 2a: The enhanced spectrum of EtOH (very nice this experiment!) should be plotted in a more extended range and compared to the Raman of EtOH. I do not agree with the attribution of the 1609 cm⁻¹ feature to water. The Raman fingerprint of water in this region is a band not a sharp peak. Please investigate better.

We thank the reviewer's suggestion. We have re-conducted the ethanol experiment to control the two AgNP-coated beads in as small as 10 nm step size for several repeated bead distance points from beads approaching to bead separating, demonstrating the excellent reproducibility of our SERS platform. The new SERS spectra of 1% ethanol were plotted in an extended range from 600 cm⁻¹ to 2000 cm⁻¹ in Fig.2 and compared with the spontaneous Raman spectrum of pure ethanol in Fig. 5S, which are in excellent agreements. We have conducted the control experiment with the brand-new ethanol sample to verify that the 1609 cm⁻¹ feature came from impurity of the old ethanol stock. Using the brand-new ethanol sample, all the Raman characters of ethanol are consistent with literature data.

When re-conducting the ethanol experiment, we performed the spectroscopic scan for individual AgNP-coated bead at the Raman excitation spot to exclude intra-bead hotspots, generating the time series spectra in Fig. 5S. At the beginning, we trapped the two AgNP-coated beads at a distance of 900 nm, generating the blank spectra similar to the spontaneous Raman of 1% ethanol in Fig. 5S. When two beads were gradually approached to less than 100 nm, the SERS signal emerged and enhanced, indicating the formation of SERS active hotspot. In particular, we measured the SERS spectra at the distance of 100 nm, 70 nm, 50 nm, 30 nm, 20 nm, 10 nm and 0 nm during the beads approaching process. At the distance of 100 nm, the observed peak at 886 cm⁻¹ was attributed to the C-C stretching of CH₃CH₂OH. As the two beads approached closer, the other characteristic peaks at 1054 cm⁻¹ (C-O stretching), 1094 cm⁻¹ (CH₃ rocking), 1280 cm⁻¹ (CH₂ deformation) and 1458 cm⁻¹ (CH₃ asymmetric bending) appeared, which were in good agreement with the spontaneous Raman spectra of pure ethanol in Fig. 5S. The maximum enhancement was achieved at 0 nm. Then we separated the two beads at the distance of 20 nm, 50 nm, 70 nm, 100 nm, and 600 nm, showing the decrease and the

disappearance of ethanol signal. The bead distance of 20 nm, 50 nm, 70 nm, and 100 nm were repeated from beads approaching to bead separating. At the same beads distance, the signal intensities obtained in beads separating was comparable to those obtained in beads approaching. This demonstrated the precise controllability and excellent reproducibility of our SERS probe. Besides, as the background, the spontaneous Raman spectra of 1% ethanol solution with 1 s acquisition time were recorded repeatedly, showing a broad water band at $\sim 1640\text{ cm}^{-1}$ in Fig. 1R. Whereas the old 1609 cm^{-1} peak was not observed in our new experiments using the brand-new ethanol sample, so we suspected that it might come from impurity or contamination. Presented in the revised manuscript, all the Raman characters of the brand-new ethanol sample are consistent with literature data.

Revised Fig.2 with all new experimental data from the measurements of ethanol:

Figure 2. Creation and adjustment of the dynamic hotspot with in situ SERS measurements of 1% ethanol aqueous solution. **a** The real-time camera images of the two AgNP-coated beads trapped at different distance. **b** SERS spectra of 1% ethanol aqueous solution with 1 s acquisition time when the two AgNP-coated beads approaching. **c** The intensity of the ethanol characteristic peak at 1458 cm^{-1} as a function of the distance between the two AgNP-coated beads from beads approaching to beads separating. The reversible bead positions are illustrated as inset. **d** SERS spectra of 1% ethanol aqueous solution with 1 s acquisition time when the two AgNP-coated beads separating.”

Revised result and discussion on the ethanol experiment in the main text:

“As a proof of concept, the dynamic SERS probe was created and adjusted between the two AgNP-coated beads on the optical tweezers-coupled Raman microscope to detect the SERS

signal of 1% ethanol aqueous solution. Two AgNP-coated beads were trapped at different distance reversibly, meanwhile the real-time camera images (Fig. 2a) and the corresponding SERS spectra with 1 s acquisition time (Fig. 2b and 2d) were recorded. At the beginning, the two beads were separated away by 900 nm on the two sides of the Raman excitation spot to acquire the blank spectrum in Fig. 2b (black), which is the same as spontaneous Raman of 1% ethanol in Fig 5S showing no apparent signal. The two beads were approached to decrease the distance below 100 nm, while the Raman signal gradually emerged and enhanced, indicating the formation of the SERS active gap at the Raman excitation spot. The characteristic peaks at 886 cm^{-1} (C-C stretching), 1054 cm^{-1} (C-O stretching), 1094 cm^{-1} (CH_3 rocking), 1280 cm^{-1} (CH_2 deformation) and 1458 cm^{-1} (CH_3 asymmetric bending)⁴⁹ are consistent with the spontaneous Raman of pure ethanol shown in Fig. 5S. To confirm the SERS enhancement effect from the hotspot between the two approached AgNP-coated beads, the entire surface of individual AgNP-coated beads was scanned in time-series measurements. The blank spectra of the single AgNP-coated bead trapped at the Raman excitation spot before and after the hotspot creation in Fig. 6S imply no intra-bead hotspots. It is worth noting that the spectral features were boosted up during beads approaching in a 10 nm step-size from 30 nm to 0 nm, indicating that the SERS active interparticle gap was under precise control and the slight adjustment resulted in the dramatic signal enhancement. In the reverse moving direction, the SERS signal gradually decreased when the two AgNP-coated beads were separated further from 20 nm to 100 nm, shown in Fig. 2d. At the same beads distance, the signal intensities obtained in beads separating are comparable to those obtained in beads approaching. Fig. 2c demonstrates the intensity of the ethanol characteristic peak at 1458 cm^{-1} as a function of the distance between the two trapped AgNP-coated beads, obtained from the trapping laser position and the brightfield image analysis. It is clear that the vibrational features of ethanol were emerged, enhanced, decreased, and vanished in inversely proportional relationship to the beads distance, under the precise control by optical tweezers with excellent flexibility and reproducibility.”

The newly added Fig 5S to compare the Raman and the SERS of ethanol:

Figure 5S. The comparison of the spontaneous Raman spectra and the SERS spectra of ethanol solutions. The SERS spectra of 1% ethanol at 0 nm nanogap (1 s, blue) was in good agreement with the spontaneous Raman spectra of pure ethanol (10 s, green). No obvious peaks were observed in both SERS spectra of 1% ethanol when the Raman excitation spot was illuminated on one bead (1 s, purple) or two beads were separated away by 900 nm (1 s, cyan), which were similar to the blank spontaneous Raman spectra of 1% ethanol (10 s, black), indicating no intra-bead hotspots. All the presented spectra were recorded directly upon the subtraction of the spontaneous Raman spectrum of 1% ethanol solution as background.”

The reference of the background for spectral subtraction:

Figure 1R. Typical Raman spectra of 1% ethanol solution with 1 s acquisition time as background, showing a broad water band at $\sim 1640 \text{ cm}^{-1}$.

4. - Force measured on the microbeads (figure 3b, page 9). It is not clear what the authors measure and how. Please clarify this important aspect.

We have conducted new complementary experiments and revised the figure to illustrate the force measurement on the trapped AgNP-coated beads. In the experiments, the AgNP-coated bead was confined in the optical trap in harmonic potentials as a force and displacement sensor, which was monitored by the position sensitive detector (PSD) shown in Fig. 1S. When the two AgNP-coated beads were separated, the AgNP-coated beads located at the center of the optical traps (Fig. 7S a) and there was no force between them (Fig. 7S b). When the two AgNP-coated beads were pushed at the bead distance of 0 nm, the AgNP-coated beads were in contact to displace from the center of the optical traps (Fig. 7S c). There was detectable force at 40 pN between the two trapped AgNP-coated beads (Fig. 7S b), which was derived from the displacements of the AgNP-coated beads in optical traps. The detectable force between the two trapped beads help us to confirm the contact of them at 0 nm distance and ensure the stability of the interparticle hotspot at the maximum SERS enhancement. However, when we conducted the protein characterizations, the two AgNP-coated beads were trapped at the distance of 20 nm without interparticle force.

The force detection system illustrated in Fig. 1S:

Figure 1S. Instrumental layout for the optical tweezers coupled Raman spectroscopic experimental platform. Abbreviations: Laser 1, 1064 nm Ytterbium linearly polarized CW laser (IPG Photonics); PBS, polarizing beamsplitter cube; L1, L2, lens; F1, filter; CMOS, complementary metal oxide semiconductor camera; L3, objective (Nikon, 60X 1.2 N.A.); S, sample holder on two-axis motorized translation micro-stage; L4, condenser (Leica, 60X 1.4 N. A.); LED, 850 nm infrared light emitting diode; L5, lens; F2, F3, filter; PSD, position sensitive detector; Laser 2, 532 nm fiber laser (CNI); NF, notch filter (Thorlab); L6, lens;

Spectrometer, IsoPlane SCT-320, 1200 lines/mm (Princeton Instrument); CCD, liquid nitrogen-cooled charge-coupled device camera (Princeton Instrument).”

Newly added Fig. 7S to illustrate the force measurement between the two AgNP-coated beads:

Figure 7S. **a, c:** Illustrations of the force measurement on the trapped AgNP-coated beads by the optical tweezers and **b:** time traces of the force detected on the optical tweezers when two AgNP-coated beads were separated and touched. When two trapped AgNP-coated beads were pushed at the bead distance of 0 nm, the force has increased rapidly from 0 pN to 40 pN, indicating the contact of two AgNP-coated beads to ensure the stability of the interparticle hotspot at the maximum SERS enhancement.”

The revised main text on the force measurement between the two trapped AgNP-coated beads: “Furthermore, the two trapped beads were pushed at a distance of 0 nm to maximize the SERS enhancement and generate the detectable force between them to confirm the contact of two beads (Fig. 7S) and ensure the stability of the SERS active interparticle gap (Fig. 6S).”

5. - Raman spectrum and SERS spectra of Hgb:

- The high intensity spectrum in Fig. 3a looks like amorphous carbon, indicating some molecular damage. Same holds for the spectra in Fig 5S.
- The spectra shown in Fig. 3c look quite different from those in d and in Fig 4S. Why? Unless this issue is afforded, it is difficult to make any assessment on the state of the molecule.
- The Raman spectrum of Hgb does not typically show the broad bands in the 1300 and 1600 region (see e.g. Wood et al. DOI: 10.1016/B978-0-12-818610-7.00013-X or Casella et al DOI 10.1016/j.saa.2011.03.048). The origin of this intense signals should be assessed. Measurements as a function of power could help.

We appreciate the reviewer’s comments and suggestions. We have performed the power-dependent SERS measurements and verified that hemoglobin would be damaged easily under high laser power to generate the strong and broad bands in the 1300 and 1600 cm^{-1} region,

which are attributed to amorphous carbon. The old results were acquired under high laser power (25 mW, 100%), thus the signal of amorphous carbon contributed significantly to the high intensity spectrum in the old Fig. 3a when the two beads were approached closely and the high intensity spectra in the old Fig. 5S when the laser excitation time was long (20 seconds). Some spectral features of hemoglobin were overwhelmed by the strong amorphous carbon signal in several SERS spectra in the old Fig. 3d and in the old Fig. 4S, deviated from the Raman and SERS spectra in the old Fig. 3c. To avoid the molecular damage and better resolve the protein information, we have re-conducted all the protein characterizations under gentle laser power (2.5 mW, 10%) and short exposure time (1 second). The new SERS spectra of hemoglobin show clear vibrational peaks with refined details, which well match the spectra reported by Wood et al. and indicate the native states of hemoglobin in terms of the oxidation state, the coordination number, and the spin state of hemes in hemoglobin.

Specifically, we have performed the power-dependent SERS measurements of 100 nM hemoglobin solution under different detection power (5 % to 100 %) at the bead distance of 20 nm, as illustrated in the new Fig.2R. When the laser power was higher than 10% (2.5 mW), the subtle spectral features of hemoglobin were overwhelmed by the intense characteristic bands arisen from amorphous carbon at $\sim 1370\text{ cm}^{-1}$ and $\sim 1580\text{ cm}^{-1}$, suggesting the damage of proteins. (doi.org/10.1103/PhysRevB.61.14095) Similar trends were found along with the decrease of bead distance or the elongation of acquisition time. Hence, the emergence of the two broad bands at $\sim 1370\text{ cm}^{-1}$ and $\sim 1580\text{ cm}^{-1}$ in the old version of Fig. 3 T5, Fig. 4S and Fig. 5S b is most likely due to the formation for amorphous carbon upon the molecular damage at high laser power.

The newly added power-dependent SERS measurements:

Figure 2R. SERS spectra of 100 nM hemoglobin solution under different detection power (5 % to 100 %) and repeated measurements under 10% (2.5 mW) laser power.

Thanks to the reviewer's suggestion, the lower laser power enables us to approach the two AgNP-coated beads closer to improve the detection limit while avoid molecular damage. We re-conducted all the SERS measurement on 100 nM hemoglobin and 1 μ M alpha-synuclein under an optimal condition at 20 nm bead distance, 10% excitation power and 1 s acquisition time. In the new version of Fig.3 and Fig.5, more spectral details could be resolved without the interference of the amorphous carbon signal. In particular, we compared our spectra and the literature spectra of hemoglobin in Fig. 3R and listed all the peaks assignments in Table. 1S. Since certain vibrational modes are sensitive to the excitation wavelength, there would be slight shifts or variations under the 532 nm excitation wavelength on our SERS platform compared to the literatures using 488 nm, 514 nm, 568 nm, 633 nm, and 785 nm wavelengths. The enhancement selectivity of heme molecules by a wide range of excitation wavelengths has been well discussed in the recommended papers (Wood et al. DOI: 10.1016/B978-0-12-818610-7.00013-X and Casella et al DOI 10.1016/j.saa.2011.03.048). Spin and oxidation marker bands such as ν_2 , ν_3 , ν_4 , and ν_{10} might vary a little bit, which are mainly determined by the source of hemoglobin and the solution environment. Overall, almost all peaks of our Raman and SERS spectra of hemoglobin are in good agreement with literatures. (Wood et al. DOI: 10.1016/B978-0-12-818610-7.00013-X and Casella et al DOI 10.1016/j.saa.2011.03.048), and (Xu, Li-Jia, et al. DOI: 10.1021/ac403974n). Our SERS spectrum matches this reference the best, due to the same excitation wavelength as 532 nm. (Xu, Li-Jia, et al. DOI: 10.1021/ac403974n) The vibrational frequencies in the SERS spectra of hemoglobin are assigned to the protein native states. Moreover, the high similarity among the SERS spectra of 100 nM hemoglobin in the new version of Fig. 3c and 3d demonstrates the reproducibility of this SERS detection window.

The comparison of our new spectra and the literature spectra in Fig.3R:

Figure 3R. The comparison of our spectra of hemoglobin under 532 nm Raman excitation wavelength and the representative spectra of hemoglobin from literature using different Raman excitation wavelengths. Blue bands highlight the peaks that are consistent between our spectra and the literature spectra. Ref: **b** Xu et al. DOI: 10.1021/ac403974n. **c** Wood et al. DOI: 10.1016/B978-0-12-818610-7.00013-X. **d** Casella et al DOI 10.1016/j.saa.2011.03.048

Newly added Table 1S for all the vibrational band assignments of hemoglobin:

Table 1S. The vibrational bands assignment of hemoglobin.

Raman Shift (cm⁻¹)	Assignment*
755	ν_{15} , ν (pyr breathing)
1006	ν_{45} , ν (C $_{\alpha}$ -C $_1$) _{asym}
1089	δ (=CH $_2$) _{asym}
1128	ν_{22} , ν (C $_{\alpha}$ -N)
1177	ν_{30} , ν (pyr half-ring) _{asym}
1232	ν_{13} , δ (C $_m$ -H)
1311	δ (CH=)
1346 for Fe(II)	
1376 for Fe(III)	ν_4 , ν (pyr half-ring) _{sym}
1404	ν_{29} , ν (pyr quarter-ring) _{sym}
1440	δ_s (=CH $_2$)
1508 for 6cLS	ν_3 , ν (C $_{\alpha}$ -C $_m$) _{sym}
1568 for 6cHS	
1594 for 6cLS	ν_2 , ν (C $_{\beta}$ -C $_{\beta}$)
1640	ν_{10} , ν (C $_{\alpha}$ -C $_m$) _{asym}

*Assignments are based on the studies by Hu et al.⁹, Kalaivani et al.¹⁰, Wood et al.¹¹, Casella et al.¹², and Mizutani¹³.

New version of Fig. 3 for the SERS measurements of 100 nM hemoglobin:

Figure 3. The spectroscopic characterizations of hemoglobin at its native states. **a** SERS spectra of 100 nM hemoglobin in aqueous solution when the two AgNP-coated beads were trapped at different distance (50 nm to 10 nm). **b** SERS spectra of 100 nM hemoglobin solution under different Raman excitation power (5 % to 100 %). **c** The comparison between SERS spectra of 100 nM hemoglobin solution (blue) with 1 s acquisition time and spontaneous Raman spectrum of 250 μ M hemoglobin solution with 60 s acquisition time (black). **d** 3D stacking plot of SERS spectra of 100 nM hemoglobin solution obtained from AgNP-coated beads trapped at 20 nm with 1 s acquisition time. **e** and **f** The histograms of the intensities of the hemoglobin characteristic peaks at 1376 cm^{-1} (mean=4929.15, RSD=14.63%) and 1594 cm^{-1} (mean=5391.41, RSD=16.35%) across the 50 SERS spectra in d, respectively.”

Revised main text on the SERS measurements of 100 nM hemoglobin:

“Under the visualization and manipulation by optical tweezers, the two AgNP-coated beads were approached at incremental distance and gradually adjusted Raman excitation power to optimize the SERS signal of 100 nM hemoglobin in aqueous solution in Fig. 3a and 3b. When the beads distance was smaller than 20 nm and the laser power was larger than 10% (2.5 mW), the subtle spectral features were overwhelmed by the intense characteristic peaks arisen from amorphous carbon ($\sim 1370 \text{ cm}^{-1}$ and $\sim 1580 \text{ cm}^{-1}$)⁵², suggesting molecular damages. Thus, the SERS detection window was set under the bead distance at 20 nm and the Raman excitation power at 10% with 1 s acquisition time. It is showcasing the great adjustability of the optical tweezers-coupled Raman spectroscopy to preserve the protein native states.

To examine if the hemoglobin retains native, five SERS spectra of 100 nM hemoglobin with 1 s acquisition time (blue) were compared to the spontaneous Raman spectrum of 250 μ M hemoglobin with 60 s acquisition time (black) in Fig. 3c. The identical frequencies between the SERS spectra and the Raman spectrum of hemoglobin indicates the similar protein states in

measurements³. As the oxidation state marker bands, the subtle peaks at 1346 cm^{-1} and 1376 cm^{-1} among these spectra are distinctly similar, attributed to the ferrous state and the ferric state⁵³, respectively. Vibrational bands at 1568 cm^{-1} and 1594 cm^{-1} corresponding to 6-coordinated high-spin heme and 6-coordinated low-spin heme are apparent at matching positions^{53, 54}. Bands appearing at 1089 cm^{-1} , 1311 cm^{-1} and 1440 cm^{-1} , assigned to the vinyl group deformation in the porphyrin ring of the heme center, are in good agreements.⁵⁵ Whereas, the marker bands of 5-coordinated high-spin heme at 1494 cm^{-1} and 1572 cm^{-1} ^{53, 54} representing the non-native state from the perturbation of metal surface, were not observed. All peak assignments of hemoglobin are listed in Table. 1S^{53, 55, 56, 57, 58}. It is clear that the SERS probe created by two AgNP-coated beads well preserved the oxidation state, the coordination number, and the spin state of hemes in hemoglobin, which are closely linked to the native structure and function of hemoglobin⁵⁴. Furthermore, the high similarity of these SERS spectra demonstrates the reproducibility of this SERS platform.”

The old version of Fig. 3a, c, d, Fig. 4S and Fig. 5S b for reference:

6. -- A very interesting range is from 200 to 800 cm^{-1} (Mizutani, DOI: 10.1016/B978-0-12-818610-7.00016-5). Can the authors show what happens also here ?

We have acquired the SERS spectrum of 100 nM hemoglobin solution in the low frequency region when we re-conducted the protein characterizations at the Raman excitation wavelength of 532 nm.

The vibrations of hemoglobin in the low frequency region are sensitive to the movement of heme iron, containing out-of-plane modes of porphyrin ring, substituent modes, and iron–ligand modes as well as in-plane modes. (Casella et al, DOI: 10.1016/j.saa.2011.03.048 and Mizutani, DOI: 10.1016/B978-0-12-818610-7.00016-5) Our result was plotted and compared to the literature spectra in Fig. 4R a. The SERS spectrum of 100 nM hemoglobin solution demonstrates the remarkable peak at $\sim 760 \text{ cm}^{-1}$ assigned to the ν_{15} , pyrrole breathing mode, which is consistent with the suggested reference (Mizutani, DOI: 10.1016/B978-0-12-818610-7.00016-5) and other references. (Wool et al., DOI: 10.1021/ja038691x and Casella et al., DOI: 10.1016/j.saa.2011.03.048) The vibrations at lower frequency region, including three in-plane A_{1g} modes ν_7 (pyrrole symmetric deformation, $\sim 675 \text{ cm}^{-1}$), ν_8 (iron-N stretching, $\sim 340 \text{ cm}^{-1}$), ν_9 (symmetric deformation between C_β and C_1 , $\sim 250 \text{ cm}^{-1}$), an out-of-plane mode γ_7 (methine wagging, $\sim 305 \text{ cm}^{-1}$), a substituent mode $\delta(C_\beta C_c C_d)$ (deformation of the propionate methylene group, $\sim 365 \text{ cm}^{-1}$), an iron-ligand mode $\nu(\text{Fe-His})$ (stretching between iron and proximal Histidine, $\sim 215 \text{ cm}^{-1}$), and two iron-oxy modes (Fe- O_2 stretching at $\sim 570 \text{ cm}^{-1}$ and Fe- O_2 bending at $\sim 420 \text{ cm}^{-1}$), (Mizutani, DOI: 10.1016/b978-0-12-818610-7.00016-5 and Hu et al. DOI: 10.1021/ja962239e) are not very apparent, because our 532 nm Raman excitation is in the vicinity of Q_v band. Similar observations are also reported by Wool et al using 488 nm and 514 nm excitation wavelengths. (DOI: 10.1021/ja038691x) In comparison, the suggested reference used shorter wavelength at 442 nm to generate the resonance Raman signals to distinguish the deoxy form and the CO-bound state of hemoglobin during the CO photolysis process. (Mizutani, DOI: 10.1016/B978-0-12-818610-7.00016-5 and DOI: 10.1016/j.chemphys.2011.05.012) Furthermore, the suggested reference also demonstrates rich resonance Raman features in the 1100 to 1700 cm^{-1} region reflecting the heme in-plane vibrations to identify the states of hemoglobin. Thus, we mainly use the high frequency region to verify that our SERS platform could preserve the native states of hemoglobin in terms of the oxidation state, the coordination number, and the spin state of hemes in hemoglobin.

The SERS spectrum of hemoglobin in the low frequency region and comparison with literature:

Figure 4R. The SERS spectrum of 100 nM hemoglobin solution in the low frequency region with 1 s acquisition time and the comparison with literature. Yellow bands highlight the peaks that are consistent between our spectra and the literature spectra. Ref: **b, d** Wood et al. DOI doi.org/10.1021/ja038691x. **c** Casella et al DOI [10.1016/j.saa.2011.03.048](https://doi.org/10.1016/j.saa.2011.03.048). **e** Mizutani, DOI: [10.1016/B978-0-12-818610-7.00016-5](https://doi.org/10.1016/B978-0-12-818610-7.00016-5).

7. - Hgb reproducibility: the spectra are characterized by very sharp peaks superimposed to broad bands. I think that some careful analysis (histograms) should be carried out also on the smaller peaks since, in my opinion, any structural information is encoded also there.

We have performed additional analysis on six small peaks of the new SERS spectra of 100 nM hemoglobin solution. We plotted new histograms in Fig. 11S to analyze the peak intensities of the characteristic peaks of hemoglobin at 1640 cm^{-1} (ν_{10} , $C_{\alpha}-C_m$ asymmetric stretching), 1568 cm^{-1} (ν_2 , spin marker band for six coordinated high spin heme), 1440 cm^{-1} (vinyl deformation), 1404 cm^{-1} (ν_{29} , symmetric stretching of pyrrole quarter-ring), 1346 cm^{-1} (ν_4 , iron oxidation state marker band for ferrous heme) and 1177 cm^{-1} (ν_{30} , asymmetric stretching of pyrrole half-ring), respectively. The relative standard deviations (RSD) across 50 parallel SERS measurements for all these characteristic peaks fall within a satisfactory range of 18.5%, demonstrating the reproducibility and stability of our SERS platform. Moreover, these subtle spectral features also provide the structural information on the heme center of hemoglobin, indicating the oxidation state, the coordination number, and the spin state of hemes as native. All the peaks assignments are summarized in Table. 1S.

Newly added Fig. 11S to show the histograms for six small peaks of hemoglobin:

Figure 11S. The histogram of the intensities of the hemoglobin characteristic peaks at 1640 cm^{-1} (mean=4413.01, RSD=15.50%), 1568 cm^{-1} (mean=5139.80, RSD=18.23%), 1440 cm^{-1} (mean=3811.98, RSD=16.13%), 1404 cm^{-1} (mean=4256.20, RSD=15.74%), 1346 cm^{-1} (mean=3876.91, RSD=15.93%) and 1177 cm^{-1} (mean=3095.36, RSD=17.35%) across the 50 SERS spectra in Fig.3 d, respectively. All the RSDs fall within a range of 18.5%, demonstrating reproducibility and stability of our SERS platform.”

8. - Sensitivity. Detecting Hgb at 10 μM is a good result but it is not what is expected by SERS in 2020. Reports in the literature show that 100nM and even 1 pM are concentration ranges are achievable (in this latter case by using an optical tweezers technique) and also single molecule SERS is feasible. The authors should explore the sensitivity limits of their technique prior to publication in Nat Commun.

We thank the reviewer’s suggestion concerning the detection sensitivity of our SERS platform. We have conducted new experiments and computational simulations to verify that our SERS platform is able to reach single-molecule level sensitivity under the precise control of two AgNP-coated beads by optical tweezers. We have also re-conducted the SERS measurements of hemoglobin at 100 nM, which is lower than the physiological concentration of hemoglobin in blood (4.7 mM to 5.6 mM Wood et al. DOI: 10.1016/B978-0-12-818610-7.00013-X). The focus of our manuscript is to show the application of SERS on characterizing the native protein structures in dilute solutions, thus we measure the SERS signal from the passing-by proteins in fluid by mechanically controlling the SERS active interparticle gap of two AgNP-coated beads to minimize the interaction between proteins and metal substrates, which is different from the

optical aggregation of gold nanoparticles with hemoglobin at 1 pM concentration for the SERS detection in the previous study. (DOI: 10.3390/ma11030440)

In order to demonstrate that our SERS platform is comparable to other SERS detection approaches with the single-molecule level sensitivity, we conducted new experiments and computational simulations. Referring to *Nature Communications* (2019) 10:5321, we incubate the AgNP-coated beads with 10^{-9} M and 10^{-11} M Rhodamine B solutions for surface absorption. In the experiment, the distance between the two trapped beads was set to 0 nm and the laser power was set at 100% to maximize the SERS enhancement for ultra-sensitive detections. In Fig. 8S, the characteristic peaks of Rhodamine B (aromatic C-C stretching at ~ 1360 cm^{-1} , ~ 1565 cm^{-1} , and ~ 1650 cm^{-1}) are observed at 10^{-9} M and even at 10^{-11} M, indicating the sensitivity of our SERS platform up to the single-molecule level. Moreover, we also performed 3D-FDTD simulation to investigate the intensity and distribution of electric field within the interparticle gap, as shown in Fig. 8S. According to the fourth power-approximation of electromagnetic enhancement factor (EF), the maximum EF is as high as 10^9 , which is sufficient to empower the single-molecule level detection.

The SERS enhancement on our platform is reversibly tunable from 0 to 10^9 under the real-time visualization and precise manipulation of two AgNP-coated beads trapped by optical tweezers to accommodate various analytes, from small chemical compounds to biomacromolecules. When characterizing the protein structures, we just set the bead distance at 20 nm and measure the SERS signal from the passing-by proteins in fluid, without incubation or adsorption process prior to SERS measurements, in order to minimize the metal-protein interaction and preserve the proteins at native states. Under gentle measurement parameters, we have re-conducted the SERS measurements of hemoglobin at 100 nM with 10% laser power and 1 second accumulation time. The concentration of 100 nM hemoglobin is lower than the physiological concentration of hemoglobin in blood (4.7 mM to 5.6 mM). With single-molecule level sensitivity, our SERS platform could be further developed for different applications involving ultrasensitive and selective detection in future investigations.

The descriptions on the verification of the single-molecule level sensitivity have been added to the main content and the supplementary information:

“Furthermore, the two trapped beads were pushed at a distance of 0 nm to maximize the SERS enhancement and generate the detectable force between them to confirm the contact of two beads (Fig. 7S) and ensure the stability of the SERS active interparticle gap (Fig. 6S). It was utilized to probe 10^{-9} M and 10^{-11} M Rhodamine B, showing the Raman signatures of the aromatic C-C stretching of Rhodamine B at ~ 1360 cm^{-1} , ~ 1565 cm^{-1} , and ~ 1650 cm^{-1} in Fig.

8S⁵⁰. The theoretical simulation indicated its SERS enhancement factor up to 10^9 , which is sufficient to empower the single-molecule level detection.⁵¹ (Details are explained in Fig. 8S.)”

Figure 8S. SERS measurements and theoretical simulation for the two AgNP-coated beads trapped at 0 nm. **a:** SERS spectra of 1 nM (green) and 10 pM (blue) Rhodamine B with 30 s acquisition time. **b:** FDTD simulation of E-field distribution ($|E/E_0|^4$) in logarithm scale, where E is the amplified local field and E_0 is the incident field. Dashed circles and solid circles represented the Ag nanoparticles and the silica beads, respectively. Considering the approximation that electromagnetic enhancement factor (EF) can be expressed as $|E/E_0|^4$, where E is the amplified local field and E_0 is the incident field¹, the maximum EF can reach as high as 10^9 as indicated by the presence of the dark-red spots.”

“SERS measurements of Rhodamine B at ultra-low concentration

AgNP-coated beads were incubated in ultra-dilute Rhodamine B solution (10^{-9} M and 10^{-11} M, respectively) for 3 h. After the surface absorption of Rhodamine B, two AgNP-coated beads were trapped and approached at 0 nm under the excitation power of 25 mW to achieve the maximum enhancement for the SERS measurements. In Fig 8S, the characteristic peaks of Rhodamine B (aromatic C-C stretching at ~ 1360 cm^{-1} , ~ 1565 cm^{-1} , and ~ 1650 cm^{-1}) are observed at 10^{-9} M and even at 10^{-11} M and they are consistent with those at 10^{-9} M, indicating the sensitivity of our SERS platform up to the single-molecule level².

3D-FDTD Simulation

To better understand the intensity and distribution of electric field in the vicinity of Raman excitation spot, three-dimensional finite-difference time-domain (3D-FDTD) simulation was carried out using FDTD SOLUTIONS provided by Lumerical Solutions, Inc. A simplified model

consisting of two Ag nanoparticle (70 nm) coated-silica beads ($R=0.63 \mu\text{m}$) was constructed. The gap between two AgNP-coated beads was set to 0 nm. The dielectric properties of Ag and SiO_2 were taken from Johnson&Christy database and Palik database, respectively. The refractive index of background fluid was set as 1.33. A 532 nm plane wave propagating along z-axis with polarization parallel to two AgNP-coated beads was employed as excitation source. Perfectly matched layer (PML) boundary condition was used and the mesh size in the nanogap was set as 0.5 nm to increase the accuracy of simulation. An overall mesh setting with mesh accuracy of 5 was applied for the rest region. Considering the approximation that electromagnetic enhancement factor (EF) can be expressed as $|E/E_0|^4$, where E is the amplified local field and E_0 is the incident field¹, the maximum EF can reach as high as 10^9 as indicated by the presence of dark-red spots in Fig. 8S b., which is sufficient to empower the single-molecule level detection³.”

9. - Physiological pH: this is always claimed but never measured. How does the presence of ethanol influence the structure of the molecules ?

We have measured the pH of the protein solutions before and after the SERS characterization on our platform, which was maintained as neutral. In the experiments, the protein solution was prepared in 1x PBS buffer at pH 7.4. Since we utilized optical tweezers to mechanically control the formation of hotspots, no capping agents (e.g. acidic MgSO_4) was added. This is different from the conventional nanoparticle-based SERS substrates generated from the salt-induced random aggregations. Thus, the pH of the protein solutions could maintain at pH 7.4 during the SERS measurements. In addition, optical tweezers could also induce optical aggregation of nanoparticles as another alternative to avoid the capping agent and maintain physiological pH for sensitive SERS measurements. (DOI: 10.3390/ma11030440 and DOI: 10.1038/srep26952).

To clarify the confusion on physiological pH, we have added new description to the material and method section:

“Protein samples were dissolved and prepared in $1 \times$ PBS buffer, pH 7.4 for SERS measurements. The pH was measured before and after the SERS characterizations to ensure that the protein solutions maintained at physiological pH in the experiments.”

We only used ethanol as an analyte in one experiment to acquire its vibrational signals to verify the creation of the real-time controllable SERS probe. As a proof of concept, the dynamic SERS probe was created and adjusted between two AgNP-coated beads on the optical tweezers coupled Raman microscope to detect the SERS signal of 1% ethanol aqueous solution. We have demonstrated that the vibrational features of ethanol were emerged, enhanced, decreased, and vanished in inversely proportional relationship to the beads distance in Fig. 2, under the precise

control by optical tweezers with excellent flexibility and reproducibility. However, we didn't use ethanol in other experiments. The microfluidic flow chamber was cleaned thoroughly between experiments, so there was no ethanol to influence molecular structures in other experiments. Furthermore, we have performed structural characterizations on hemoglobin, lysozyme, and bovine serum albumin (BSA) to confirm that our controllable SERS platform can preserve the protein structures at the native states without any perturbation.

To avoid confusion, we have added the new description on experimental procedures to the material and method section:

“Protein samples were dissolved and prepared in 1× PBS buffer, pH 7.4 for SERS measurements..... Ethanol was only used as the analyte in the first experiment whereas not involved in other experiments. The microfluidic flow chamber was cleaned thoroughly between experiments.”

10. - alfa syneuclin:

-- Figure 4b: Is the Raman spectrum of 2mM background subtracted ? Please comment.

-- I understand that 250 μ M is below the sensitivity threshold of Raman, but some small features are visible, together with the Raman band of water (asterisk). Please comment. Also pls compare the Raman

In the old version of Fig. 4b, the Raman spectrum of 2 mM alpha-synuclein still contains background. The Raman spectrum of 250 μ M alpha-synuclein also contains background, thus the broad Raman band of water is apparent. The small features in the Raman spectrum of 250 μ M alpha-synuclein are also visible in that of 2 mM alpha-synuclein, which do not match the vibrational signature of alpha-synuclein. Hence, they may come from the background noise. For better analysis and comparison, we cleaned the optics thoroughly and re-conducted the Raman measurements. We performed the background subtraction on the Raman spectra of 2 mM and 250 μ M alpha-synuclein to plot new Fig. 5b. Since the data quality have been significantly improved, the noisy features are eliminated in our new spectra. The Raman spectrum of 2 mM alpha-synuclein shows the vibrational features in amide III region at 1249 cm^{-1} and amide I region at 1673 cm^{-1} , which are attributed to the majority population of alpha-synuclein with disordered conformations. In comparison, only few peaks are barely observed in the Raman spectrum of 250 μ M alpha-synuclein under the zoom-in view in Fig. 5Rd, which look like the spectral features of the Raman spectrum of 2 mM alpha-synuclein at around 1249 cm^{-1} (amide III band), 1450 cm^{-1} (deformation from aliphatic residues CH_2 and CH_3), and 1673 cm^{-1} (amide I band). It is difficult to define the accurate positions and intensities of the small features of the Raman spectrum of 250 μ M alpha-synuclein. Hence, it is insufficient to use Raman spectroscopy to characterize the secondary structure of alpha-synuclein at low

concentration, due to its low cross-section of spontaneous Raman and more gentle laser power (25 mW) in our setup.

The Raman spectra of alpha-synuclein and the background spectrum for subtraction:

*Figure 5R. Spontaneous Raman spectra of 2 mM and 250 μ M alpha-synuclein solution with 10 min acquisition time. **a**: Spontaneous Raman spectra of 2 mM alpha-synuclein before and after background subtraction. **b**: Spontaneous Raman spectra of 250 μ M alpha-synuclein before and after background subtraction. **c**: Spontaneous Raman spectra of 2 mM and 250 μ M alpha-synuclein after background subtraction. **d**: Zoomed-in spontaneous Raman spectrum of 250 μ M alpha-synuclein after background subtraction and spontaneous Raman spectrum of 2 mM alpha-synuclein after background subtraction.*

New version of Fig. 5b showing the Raman spectra of alpha-synuclein after the background subtraction:

Figure 5b. Spontaneous Raman spectra of 2 mM (green) and 250 μ M (blue) alpha-synuclein solution with 10 min acquisition time.

Revised description on the Raman of alpha-synuclein in the main text:

“In Fig. 5b, the spontaneous Raman spectrum of 2 mM alpha-synuclein solution shows the vibrational features in amide III region at 1249 cm^{-1} and amide I region at 1673 cm^{-1} , which are attributed to the majority population of disordered conformations¹⁶. The protein signal of 250 μM alpha-synuclein solution in Fig. 5b is too weak to analyze, due to its low Raman cross-section and the gentler laser power (25 mW) in our setup, lower than the 800 mW laser power used in the previous Raman studies^{16, 17}.”

Newly added description on the background subtraction in the material and method section:

“All presented spectra were obtained upon the subtraction of the background accordingly and smoothed by Savitzky–Golay filter.”

The old version of Fig. 4b for reference:

11. -- SERS sensitivity. The authors show nice spectra at 10 μM . Some comparison with the SERS detection limits in the literature should be carried out and some exploration of the real limits of this technique on alpha-syneucline should be a very valuable information.

We have conducted new experiments and computational simulations to verify that our SERS platform is able to reach single-molecule level sensitivity under the precise control of two AgNP-coated beads by optical tweezers, as demonstrated in the previous sections. We have also conducted the new SERS measurements of alpha-synuclein at 10 nM, 30 nM, 100 nM, and 1 μM under the 20 nm bead distance, the 10% laser power and 1 s accumulation time. As shown in Fig. 15S, the spectral features of alpha-synuclein were still distinguishable at 100 nM, while the amide I band and the band of CH₂, CH₃ deformation were barely visible at 30 nM and 10 nM. Thus, the minimum visible concentration of alpha-synuclein aqueous solution under this experimental condition is at around 100 nM. This experimental limit of detection (LOD) of alpha-synuclein on our SERS platform is the lowest compared to the Raman and SERS detection limits of alpha-synuclein reported in the literature.

Figure 15S. SERS spectra of alpha-synuclein at 1 μ M, 100 nM, 30 nM and 10 nM obtained at the 20 nm bead distance and the 10% laser power with 1 s acquisition time on our SERS platform, showing the minimum visible concentration of alpha-synuclein aqueous solution is at around 100 nM under the current SERS experimental condition.”

Previous studies mainly use Raman spectroscopy to characterize the conformational ensembles of the heterogenous alpha-synuclein in solution phase and the structural features of alpha-synuclein amyloid fibrils. (DOI: 10.1021/ja0356176 and DOI: 10.1074/jbc.M117.812388) The SERS study on alpha-synuclein is very limited, which requires the absorption of proteins on the surface of metal nanoparticles. It was reported that using the liquid core photonic crystal fiber SERS sensor with the silver binding peptides to immobilize the alpha-synuclein, the detectable concentration could reach 10^{-4} M~ 10^{-5} M. (DOI: 10.1117/12.760117) Recently we have optimized our experimental conditions and pushed the detection limit of alpha-synuclein to 100 nM with 1 s acquisition time under the 20 nm bead distance and the 10% laser power, as shown in Fig. 15S. To the best of our knowledge, the minimum visible concentration of alpha-synuclein in solution phase at 10^{-7} M on our SERS platform is the lowest in comparison with the literature detection limits. Furthermore, the physiological concentration of alpha-synuclein is 1 μ M. The subtle spectral features of alpha-synuclein acquired at this concentration could reveal its structural details with great biological significance. Thus, we focus on the SERS characterization of 1 μ M alpha-synuclein, which is also reported for the first time. Overall, our SERS platform supports the sensitive detection of these intrinsically disordered proteins to reveal their structural features in the native states at physiological concentration.

Newly added Fig. 15S to show the experimental limit of detection (LOD) of alpha-synuclein on our SERS platform:

Figure 15S. SERS spectra of alpha-synuclein at 1 μM , 100 nM, 30 nM and 10 nM obtained at the 20 nm bead distance and the 10% laser power with 1 s acquisition time on our SERS platform, showing the minimum visible concentration of alpha-synuclein aqueous solution is at around 100 nM under the current SERS experimental condition.”

Revised main text on the new SERS characterization of 1 μM alpha-synuclein:

“In particular, the SERS study on alpha-synuclein, an IDP closely related to Parkinson’s disease^{15, 16, 17}, is scarce¹⁸.”

“The physiological concentration of alpha-synuclein at non-aggregated states is 1 μM ⁴⁷, below the detection threshold of Raman spectroscopy. Whereas the experimental limit of detection (LOD) of alpha-synuclein on our SERS platform is 100 nM, thus this sensitive SERS approach is feasible to characterize the transient species of alpha-synuclein in dilute solutions. (Details are shown in Fig. 15S and Fig. 20S.)”

“Owing to high sensitivity and stability, our SERS platform resolved the structural variations of alpha-synuclein arisen from its transient species, which is the first SERS characterization of alpha-synuclein at physiological concentration.”

12. -- Structure of the protein. I regret to say that this article does not provide any useful information on this issue. Indeed the spectra vary, and this can be a sign of conformational changes, but this is obvious and does not add valuable information. The authors are not able to drive the experimental conditions so to control and probe the structure in each case.

We have performed a new experiment to characterize 1 μM alpha-synuclein at pH 3 in Fig. 16S, converting alpha-synuclein into ordered β -sheet conformations and verify the accuracy of our structural characterization. More importantly, we have revised the introduction on alpha-synuclein to highlight the heterogeneous nature of intrinsically disordered proteins (IDPs) that usually exist in a dynamic equilibrium mixture of various structures in the aqueous solution, which make it very challenging to investigate. As a comparison, we have conducted new supplementary experiments on two well-known globular proteins: lysozyme and bovine serum albumin (BSA) that only possess the homogenous structure with small structural fluctuations in the aqueous solution. With high sensitivity, our SERS platform could characterize alpha-synuclein in dilute aqueous solution to reduce ensemble averaging and resolve its conformational variation, generating varied spectra to reveal the structural features of different species of alpha-synuclein in the heterogeneous mixture. We have revised the main texts and figures to highlight the contribution of our structural characterization on alpha-synuclein at physiological concentration in aqueous solution, maintaining its intrinsic heterogeneity with great biological significance. Specifically, since the 1 μM physiological concentration of alpha-synuclein is below the detection threshold of Raman spectroscopy, the SERS characterization of the secondary structure of the non-aggregated alpha-synuclein in solution phase at such low concentration is reported for the first time, indicating the co-existence of monomeric species and different transient species. Moreover, these varied SERS spectra obtained from the small-size sampling directly characterize the structures of the transient species of alpha-synuclein in the dynamic equilibrium mixture, which are buried under the averaging signals in the traditional bulk measurements but are crucial as the initiation point to determine the alpha-synuclein aggregation. Furthermore, our spectral analysis provides an accurate assessment on the distribution of different secondary structure components of alpha-synuclein at such low physiological concentration in the aqueous solution, superior to other experimental techniques (e.g. CD and NMR) as suggested by the first reviewer.

Figure 16S. SERS spectrum of 1 μM alpha-synuclein at pH 3 with 1 s acquisition time, showing β -sheet conformation.”

We have performed a new experiment to characterize 1 μM alpha-synuclein at pH 3 in Fig. 16S, because alpha-synuclein folds into ordered β -sheet conformations in acidic conditions. The shape of amide I band at around 1664 cm^{-1} and the amide III band at $1230\text{-}1240\text{ cm}^{-1}$ further confirm our assignment on the second type of spectra of alpha-synuclein in aqueous solution as the β -sheet structure. Although changing the experimental conditions (e.g. pH, methanol, SDS) can convert alpha-synuclein into more uniform conformations and alter the distribution of transient species, our focus is to differentiate and characterize the structures of different species of alpha-synuclein in the dynamic equilibrium mixture in aqueous solution at neutral pH, maintaining its intrinsic heterogeneity with great biological significance.

As a typical intrinsically disordered protein, alpha-synuclein lacks stable structures in aqueous solution, showing the ensemble conformation as random coil. However, it undergoes dynamic structural conversions and self-assembles into different transient species with various structures in low population, which are difficult to characterize due to the ensemble averaging in the traditional bulk measurements. (DOI: 10.1038/nature16531) The conversions between different structures of alpha-synuclein are intrinsic and dynamic, thus it is challenging to control the structure of transient species of IDPs without altering the distribution of transient species and monomeric species. Since our platform enables the sensitive and reproducible SERS characterizations, we could directly identify the structural variations arisen from the transient species of alpha-synuclein under the parallel small-size sampling SERS measurements in dilute concentration (1 μM) and short accumulation time (1 s). Characterizing the passing-by proteins in fluid at the controllable SERS hotspot without incubation or adsorption, we are able to reduce ensemble averaging to directly resolve the structural details of different species of alpha-synuclein in the dynamic equilibrium mixture. Our observation on the conformational variations of alpha-synuclein reflects the heterogeneous nature of intrinsically disordered proteins and the small-size sampling is an effective strategy to probe the structures of alpha-synuclein transient species. Hence, our method opens a new door to characterize the unique structural features of IDPs in dilute solutions, which remains a significant challenge in the biophysics community.

Newly added Fig. 4 for the SERS measurements of lysozyme:

Figure 4. The spectroscopic characterizations of lysozyme in its compact globular structure. **a** The comparison between 50 SERS spectra of 1 μM lysozyme solution with 1 s acquisition time (bottom) and the spontaneous Raman spectrum of 1 mM lysozyme solution with 5 min acquisition time (top). **b** Histogram of the Amide I band distribution of the 50 SERS spectra of 1 μM lysozyme solution in **a**, indicating the mean as 1655 cm^{-1} with 0.1% RSD. **c** Illustration of the ensemble averaging from the spontaneous Raman measurement of lysozyme in the concentrated solution and the small-size sampling from the SERS measurements of lysozyme in the dilute solution.”

Revised Fig. 5 for the SERS measurements of alpha-synuclein:

Figure 5. The spectroscopic characterizations of an intrinsically disordered protein: alpha-synuclein. **a** CD spectrum of 200 μM alpha-synuclein in aqueous solution. **b** Spontaneous Raman spectra of 2 mM (green) and 250 μM (blue) alpha-synuclein solution with 10 min acquisition time. **c** The comparison among three representative types of SERS spectra of 1 μM alpha-synuclein solution with 1 s acquisition time (blue, red, and black) and the SERS spectrum

of 250 μM alpha-synuclein solution with 5 min acquisition time (purple). d Mapping of 200 SERS spectra of 1 μM alpha-synuclein solution obtained from two AgNP-coated beads trapped at 20 nm with 1 s acquisition time. The color bar shows the normalized intensities from low (dark blue) to high (red). e Illustration of the ensemble averaging from the measurement of alpha-synuclein at high concentration with long accumulation time and the small-size sampling from the measurements of alpha-synuclein at low concentration with short accumulation time.”

Buried under the averaging signals in the bulk, the conformational variations of alpha-synuclein in the dynamic equilibrium mixture are subtle yet significant. Here, among 200 parallel small-size sampling SERS measurements on our platform, the amide I and the amide III bands of the SERS spectra of 1 μM alpha-synuclein aqueous solution exhibit prominent variations in comparison to the uniform spectral features of 1 μM lysozyme aqueous solution, indicating the co-existence of different alpha-synuclein species with diverse conformations even at such low concentration. More importantly, it directly characterized the structural features of the transient species of alpha-synuclein in the dynamic equilibrium mixture. In particular, the small portion of β -sheet containing transient species of alpha-synuclein might link to its pathological misfolding to initiate the amyloid aggregation at the very early stage. Interestingly, the small structural fluctuation of lysozyme (possessing the identical folded structure from one to another) and the large structural fluctuation of alpha-synuclein (existing in different conformations from one to another) represents the homogeneous nature of compact globular proteins and the heterogeneous nature of IDPs, respectively. Thus, the small-size sampling is an effective strategy to reduce ensemble averaging and directly probe the transient species of alpha-synuclein in the dynamic mixture (doi.org/10.1021/cr400297g), without the need to synchronize the intrinsic and heterogeneous conformational conversions for bulk measurements like the traditional pump-probe approaches. Finally, the parallel SERS measurements under the small-size sampling could still provide the ensemble conformational information on population and probability in statistics, comparable and complementary to the spectral deconvolution of the ensemble averaging spectrum in the bulk measurement, in order to resolve the distribution of the secondary structure components. Therefore, such direct identification of the structural variation of alpha-synuclein in dilute aqueous solution verifies that our sensitive SERS platform could reduce the ensemble averaging to reveal more structural details of the transient species of amyloidogenic proteins prior to the amyloid aggregation, providing profound molecular insights to understand the onset of neurodegenerative diseases.

To summarize, we have added the introduction of the heterogenous nature of intrinsically disordered proteins (e.g. alpha-synuclein) to avoid confusion, added the new experimental results of lysozyme as an example of globular proteins to compare their structural fluctuations,

added the new experimental result of 1 μM alpha-synuclein at pH 3 to characterize and verify its β -sheet structure, and revised the discussion on the spectral results of alpha-synuclein to provide more structural information and implication on its transient species in the dynamic equilibrium mixture. The revisions are shown below:

“It is also a powerful tool to characterize the dynamic ensembles of variable conformations of intrinsically disordered proteins (IDPs)^{7, 8, 9, 10}. IDPs lack stable secondary and tertiary structures as monomers in aqueous environments, but sometimes self-assemble into oligomers with various structures and further grow into amyloid fibrils¹¹, which are associated with the incurable neurodegenerative diseases¹². The dynamic conversion from monomers to oligomers is the key step in the early pathological development¹³. However, the transient nature and the low population of oligomers make it challenging to characterize their structural features, which are responsible for the cellular toxicity and the on-going amyloid aggregation¹⁴. In particular, the SERS study on alpha-synuclein, an IDP closely related to Parkinson’s disease^{15, 16, 17}, is scarce¹⁸.”

Figure 4. The spectroscopic characterizations of lysozyme in its compact globular structure. **a** The comparison between 50 SERS spectra of 1 μM lysozyme solution with 1 s acquisition time (bottom) and the spontaneous Raman spectrum of 1 mM lysozyme solution with 5 min acquisition time (top). **b** Histogram of the Amide I band distribution of the 50 SERS spectra of 1 μM lysozyme solution in **a**, indicating the mean as 1655 cm^{-1} with 0.1% RSD. **c** Illustration of the ensemble averaging from the spontaneous Raman measurement of lysozyme in the concentrated solution and the small-size sampling from the SERS measurements of lysozyme in the dilute solution.”

“Fig. 4b demonstrates the Amide I band distribution of the 50 SERS spectra of 1 μM lysozyme solution with $1655 \pm 2 \text{ cm}^{-1}$ (0.1% RSD), indicating its structural stability and homogeneity as a typical globular protein. It is worth noting that the 50 SERS spectra of 1 μM lysozyme in Fig. 4a demonstrate high similarities, due to the stability and reproducibility of this SERS probe as well as the nature of the stable, compact, and globular conformation of lysozyme.”

Figure 5. The spectroscopic characterizations of an intrinsically disordered protein: alpha-synuclein. **a** CD spectrum of 200 μM alpha-synuclein in aqueous solution. **b** Spontaneous Raman spectra of 2 mM (green) and 250 μM (blue) alpha-synuclein solution with 10 min acquisition time. **c** The comparison among three representative types of SERS spectra of 1 μM alpha-synuclein solution with 1 s acquisition time (blue, red, and black) and the SERS spectrum of 250 μM alpha-synuclein solution with 5 min acquisition time (purple). **d** Mapping of 200 SERS spectra of 1 μM alpha-synuclein solution obtained from two AgNP-coated beads trapped at 20 nm with 1 s acquisition time. The color bar shows the normalized intensities from low (dark blue) to high (red). **e** Illustration of the ensemble averaging from the measurement of alpha-synuclein at high concentration with long accumulation time and the small-size sampling from the measurements of alpha-synuclein at low concentration with short accumulation time.”

Figure 16S. SERS spectrum of 1 μM alpha-synuclein at pH 3 with 1 s acquisition time, showing β -sheet conformation.”

“As a typical IDP, alpha-synuclein undergoes intrinsic conformational conversions to form transient species in a low population at physiological concentration, existing in a dynamic equilibrium mixture to determine its amyloid aggregation at the early stage⁶⁴. With the large sample quantity and long detection time in the bulk spectroscopic measurements, the structural features of alpha-synuclein transient species are overwhelmed by the conformational ensemble. Fig. 5c demonstrates three representative types of SERS spectra of 1 μ M alpha-synuclein solution with 1 s acquisition time among the 200 parallel experiments shown in Fig. 5d. Strikingly, the amide I and the amide III bands of the SERS spectra of 1 μ M alpha-synuclein exhibit prominent variations in comparison to the uniform spectral patterns of 1 μ M lysozyme in Fig. 4, indicating the co-existence of different alpha-synuclein species with various structures at physiological concentration⁷. In Fig. 5c, the spectral characteristics of the blue SERS spectrum of 1 μ M alpha-synuclein fall in the intervals of 1650-1657 cm^{-1} (amide I bands) and 1270-1300 cm^{-1} (amide III bands), indicating the alpha-synuclein species in α -helix structure^{16, 65}. The red SERS spectrum of 1 μ M alpha-synuclein shows the amide I band at around 1662-1665 cm^{-1} and the amide III band at 1230-1240 cm^{-1} , which are associated with β -sheet structure^{16, 17, 65}. This assignment is further confirmed by the SERS spectrum of 1 μ M alpha-synuclein at pH 3 in Fig. 16S, since alpha-synuclein folds into ordered β -sheet conformations in acidic conditions. The amide I band at around 1671 cm^{-1} and the amide III band at 1240-1250 cm^{-1} from the black SERS spectrum of 1 μ M alpha-synuclein are attributed to random coil structure^{16, 65, 66}, consistent with its spontaneous Raman spectrum. While the vibrational fingerprints from Phe (1006 cm^{-1}) and deformation from aliphatic residues CH_2 and CH_3 (1450 cm^{-1}) of alpha-synuclein¹⁶ are still uniform across these SERS spectra of 1 μ M alpha-synuclein, since they are insensitive to the change of protein conformations. All the peak assignments of the 1 μ M alpha-synuclein SERS spectra are summarized in Table 2S¹⁶, since the subtle spectral features of alpha-synuclein acquired at physiological concentration could reveal the structural details of its transient species with great biological significance. In particular, the direct characterization of the β -sheet containing oligomers among the unstructured monomers of alpha-synuclein might provide new insight to the pathological aggregation of alpha-synuclein at the very early stage as this conformation is involved prior to fibrillation^{67, 68}.”

13. -- Some comparison between the a 10uM Raman and the SERS should be given.

We have compared the 2 mM Raman, the 1 μ M Raman and the 1 μ M SERS of alpha-synuclein in Fig. 20S. Since we have optimized the measurement parameters to characterize alpha-synuclein at 1 μ M in the revised manuscript, we plotted the new 1 μ M SERS instead of the old 10 μ M SERS. We have also conducted the new complementary experiment to acquire the 1 μ M Raman and the 2 mM Raman of alpha-synuclein for comparison.

“*Figure 20S. The comparison of the SERS spectrum of 1 μ M alpha-synuclein solution (1 s acquisition time) and the spontaneous Raman spectra of 1 μ M and 2 mM alpha-synuclein solution (10 min acquisition time).*”

As shown in Fig. 20S, there is no apparent protein signal in the Raman spectrum of 1 μ M alpha-synuclein (red spectrum), due to its low Raman cross-section and the gentler laser power (25 mW) in our setup. While we could obtain the SERS spectrum of 1 μ M alpha-synuclein solution with subtle vibrational features (blue spectrum). These vibrational frequencies of the SERS spectrum of 1 μ M alpha-synuclein are in good agreement with those of the spontaneous Raman spectrum of 2 mM alpha-synuclein (black spectrum), highlighted in blue bars in Fig. 20S. In particular, the vibrational fingerprints from Phe (1006 cm^{-1}) and deformation from aliphatic residues CH_2 and CH_3 (1450 cm^{-1}) of alpha-synuclein are identical at the matching peak positions. The 1671 cm^{-1} amide I band and the 1240-1250 cm^{-1} amide III band from the SERS spectrum of 1 μ M alpha-synuclein are attributed to random coil structure, which are consistent with the 2 mM alpha-synuclein Raman spectrum. Therefore, the comparison between the 1 μ M alpha-synuclein Raman and the 1 μ M alpha-synuclein SERS verifies the high sensitivity of our SERS platform.

Newly added Fig. 20S to compare the 2 mM alpha-synuclein Raman spectrum, the 1 μ M alpha-synuclein Raman spectrum and the 1 μ M alpha-synuclein SERS spectrum:

Figure 20S. The comparison of the SERS spectrum of 1 μM alpha-synuclein solution (1 s acquisition time) and the spontaneous Raman spectra of 1 μM and 2 mM alpha-synuclein solution (10 min acquisition time).”

Revised main text on the comparison between the 1 μM Raman and the 1 μM SERS of alpha-synuclein:

“The physiological concentration of alpha-synuclein at non-aggregated states is 1 μM ⁴⁷, below the detection threshold of Raman spectroscopy. Whereas the experimental limit of detection (LOD) of alpha-synuclein on our SERS platform is 100 nM, thus this sensitive SERS approach is feasible to characterize the transient species of alpha-synuclein in dilute solutions. (Details are shown in Fig. 15S and Fig. 20S.)”

14. Minor concerns

- Please change "at the physiological concentration" with "at physiological concentration" whenever it applies.

We thank the reviewer’s wording suggestion. We have changed all the "at the physiological concentration" to "at physiological concentration".

15. Page 3, second paragraph. "This dynamic SERS detection has great tunability in the microfluidic flow ..." Please specify better this point. What is versatile in the microfluidic cell ?

We have revised this sentence to clarify that we are able to adjust the bead distance and the flow rate of the protein solution in the SERS measurements, in order to preserve the native states and conformations of proteins. It has been reported that a moderately high flow rate can basically eliminate the disturbance of protein structures while maintain a high SERS enhancement factor in the microfluidic SERS detection system. (DOI: 10.1021/ja8006337)

Revised main text on the microfluidic setting:

“This dynamic and tunable SERS window was placed in a microfluidic flow chamber to detect the passing-by proteins. The flow rate of the protein solution was fine-tuned to minimize the interaction between protein and AgNP-coated beads, in order to preserve their native states and conformations.”

Reviewer #3 (Remarks to the Author):

X. Dai and co-authors report on a new SERS probe for protein spectroscopy. The probe is based on optical tweezers where two laser beams are used to control the gap distance between two Ag decorated nanoparticles.

The idea to use the double laser beam to control the trapping and the SERS enhancement is nice and, accordingly to the results presented in the manuscript, it seems to work.

Anyway, in my opinion the manuscript in the present form is not suitable for nature communications (maybe it can be for a sister mid-impact journal).

1. First of all the authors discuss the potential application of SERS in single molecule spectroscopy. This is a well known research field and recently some important results have been published also in Nat. Comm (see for instance Nat. Commun. 9, 1733 (2018) and NATURE COMMUNICATIONS (2019) 10:5321). In particular the demonstration of single molecule protein spectroscopy in SERS platform integrated with optical / plasmonic tweezers has been reported in doi.org/10.1002/anie.202000489.

The results presented in the submitted manuscript is rather far from these state-of-the-art methods. In fact no single molecule sensitivity is discussed. Moreover it's not clear if the platform can reach this sensitivity. Considering the calculated EF it seems not the case.

We appreciate the reviewer for listing the application of single-molecule SERS on DNA/protein sequencing (the primary structure) and pointing out an evaluation criterion on the detection

sensitivity of our SERS platform to compare with other SERS methods. Thus, we have conducted new experiments and computational simulations to verify that our optical tweezers-coupled SERS platform is able to reach single-molecule level sensitivity, which is comparable to these state-of-the-art methods. Furthermore, the SERS enhancement on our platform is reversibly tunable and reproducible under the real-time visualization and precise manipulation of two AgNP-coated beads trapped by optical tweezers to accommodate various analytes, from small chemical compounds to biomacromolecules. More importantly, the focus of our manuscript is to show the application of SERS on characterizing the secondary structure of proteins at native states without the absorption onto SERS substrates in dilute aqueous solutions, especially the structural variation of intrinsically disordered proteins arisen from the transient species with different structures at physiological concentration.

Indeed, advancements of single-molecule SERS have been achieved in various applications. Mechanical manipulations, in particular, optical/plasmonic trapping techniques, can create the single hotspot at the designated location with extraordinary spectral and spatial sensitivity. For example, the suggested references have employed electro-plasmonic tweezers to create the hotspot by trapping a gold nanourchin (AuNU) in a plasmonic nanohole at about 5 nm to discriminate the sequence of DNA and protein. (NATURE COMMUNICATIONS (2019) 10:5321 and doi.org/10.1002/anie.202000489) We have revised the introduction for a more comprehensive literature review and replenished the missing citations:

“To address this dilemma, mechanical manipulations are desirable alternatives to create hotspots dynamically at the designated location^{7, 30, 31}.”

Transferring the advantages of these remarkable works to a different application, we have developed the micrometer-size AgNP-coated beads for better visualization and manipulation to create the tunable SERS hotspot on our optical tweezers-coupled SERS platform, which makes it suitable for the characterization of the secondary structures of proteins in aqueous solution with high accuracy and at low concentration. We used a relatively large nanogap (20 nm) and gentled excitation laser (2.5 mW) to minimize the interaction between the proteins and the metallic nanogap without incubation or adsorption process prior to SERS measurement, in order to preserve the protein conformations at native states. With excellent sensitivity and reproducibility, we successfully identified the structural features of the transient species of alpha-synuclein among its unstructured monomers at physiological concentration. To the best of our knowledge, the SERS characterization of alpha-synuclein at such low concentration is reported for the first time.

To verify the single-molecule level sensitivity, we conducted new experiments and computational simulations on our SERS platform. Referring to NATURE

COMMUNICATIONS (2019) 10:5321, we incubate the AgNP-coated beads with 10^{-9} M and 10^{-11} M Rhodamine B solutions for surface absorption. In the experiment, the distance between the two trapped beads was set to 0 nm and the laser power was set at 100% (25 mW) in order to achieve the maximum enhancement. In Fig 8S, the characteristic peaks of Rhodamine B (aromatic C-C stretching at ~ 1360 cm^{-1} , ~ 1565 cm^{-1} , and ~ 1650 cm^{-1}) are observed at 10^{-9} M and even at 10^{-11} M, indicating the sensitivity of our SERS platform up to the single-molecule level. Moreover, we also performed 3D-FDTD simulation to investigate the intensity and distribution of electric field within the interparticle gap, as shown in Fig. 8S. According to the fourth power-approximation of electromagnetic enhancement factor (EF), the maximum EF is as high as 10^9 , which is sufficient to empower the single-molecule level detection.

The descriptions on the verification of the single-molecule level sensitivity have been added to the main content and the supplementary information.

“Furthermore, the two trapped beads were pushed at a distance of 0 nm to maximize the SERS enhancement and generate the detectable force between them to confirm the contact of two beads (Fig. 7S) and ensure the stability of the SERS active interparticle gap (Fig. 6S). It was utilized to probe 10^{-9} M and 10^{-11} M Rhodamine B, showing the Raman signatures of the aromatic C-C stretching of Rhodamine B at ~ 1360 cm^{-1} , ~ 1565 cm^{-1} , and ~ 1650 cm^{-1} in Fig. 8S⁵⁰. The theoretical simulation indicated its SERS enhancement factor up to 10^9 , which is sufficient to empower the single-molecule level detection.⁵¹ (Details are explained in Fig. 8S.)”

Figure 8S. SERS measurements and theoretical simulation for the two AgNP-coated beads trapped at 0 nm. **a:** SERS spectra of 1 nM (green) and 10 pM (blue) Rhodamine B with 30 s acquisition time. **b:** FDTD simulation of E-field distribution ($|E/E_0|^4$) in logarithm scale, where E is the amplified local field and E_0 is the incident field. Dashed circles and solid circles

represented the Ag nanoparticles and the silica beads, respectively. Considering the approximation that electromagnetic enhancement factor (EF) can be expressed as $|E/E_0|^4$, where E is the amplified local field and E_0 is the incident field¹, the maximum EF can reach as high as 10^9 as indicated by the presence of the dark-red spots.”

“**SERS measurements of Rhodamine B at ultra-low concentration**

AgNP-coated beads were incubated in ultra-dilute Rhodamine B solution (10^{-9} M and 10^{-11} M, respectively) for 3 h. After the surface absorption of Rhodamine B, two AgNP-coated beads were trapped and approached at 0 nm under the excitation power of 25 mW to achieve the maximum enhancement for the SERS measurements. In Fig 8S, the characteristic peaks of Rhodamine B (aromatic C-C stretching at ~ 1360 cm^{-1} , ~ 1565 cm^{-1} , and ~ 1650 cm^{-1}) are observed at 10^{-9} M and even at 10^{-11} M and they are consistent with those at 10^{-9} M, indicating the sensitivity of our SERS platform up to the single-molecule level².

3D-FDTD Simulation

To better understand the intensity and distribution of electric field in the vicinity of Raman excitation spot, three-dimensional finite-difference time-domain (3D-FDTD) simulation was carried out using FDTD SOLUTIONS provided by Lumerical Solutions, Inc. A simplified model consisting of two Ag nanoparticle (70 nm) coated-silica beads ($R=0.63$ μm) was constructed. The gap between two AgNP-coated beads was set to 0 nm. The dielectric properties of Ag and SiO_2 were taken from Johnson&Christy database and Palik database, respectively. The refractive index of background fluid was set as 1.33. A 532 nm plane wave propagating along z-axis with polarization parallel to two AgNP-coated beads was employed as excitation source. Perfectly matched layer (PML) boundary condition was used and the mesh size in the nanogap was set as 0.5 nm to increase the accuracy of simulation. An overall mesh setting with mesh accuracy of 5 was applied for the rest region. Considering the approximation that electromagnetic enhancement factor (EF) can be expressed as $|E/E_0|^4$, where E is the amplified local field and E_0 is the incident field¹, the maximum EF can reach as high as 10^9 as indicated by the presence of dark-red spots in Fig. 8S b,, which is sufficient to empower the single-molecule level detection³.”

2. The manuscript then illustrates three different experiments on SERS with the proposed optical tweezers. The first one is a very simple Raman spectroscopy of EtOH and it clearly demonstrates that they can control the distance between two beads in order to achieve high EF. Anyway, the authors claim some sub-nm control of the distance but no demonstration of that is reported. This is a very crucial point.

We have conducted new complementary experiments to demonstrate the sub-nanometer control of the AgNP-coated beads on our SERS platform. Specifically, we utilized the optical tweezers to trap and move an AgNP-coated bead in the well-defined step size, e.g. 3 nm, 2 nm, 1 nm, and 0.5 nm. Fig. 4S shows the observed bead distance increment as a function of the step number (5 steps in total). Each distance data point was the average position of the trapped AgNP-coated bead in 1 second against its Brownian motion. (1 second is also the shortest spectroscopic accumulation time in this work.) These results demonstrate that the AgNP-coated bead could be moved to the designated distance after each optical tweezer movement, even with the step size as small as 0.5 nm. It is also supported by literatures that the spatial resolution of optical tweezers is typically 0.2–0.4 nm. (doi: 10.1038/nature04268, doi: 10.1073/pnas.0603342103, doi: 10.1038/nmeth.1574, and doi: 10.1063/1.4752190) Thus, the coupled optical tweezers in our SERS platform can manipulate the AgNP-coated bead with sub-nanometer spatial resolution.

With the precise control of the AgNP-coated beads, our SERS platform can achieve not only the high SERS enhancements but also the tunable, reversible, and reproducible SERS enhancements, which make it suitable for the characterization of various analysts, from small chemical compounds to biomacromolecules.

Newly added Fig. 4S demonstrating the sub-nanometer spatial resolution of the optical tweezers on our SERS platform:

Figure 4S. Incremental movements of an AgNP-coated bead in different step size (5 steps in total). a: 3 nm step size. b: 2 nm step size. c: 1 nm step size. d: 0.5 nm step size. After each movement operation by optical tweezers, the distance data points of the trapped AgNP-coated

bead within 1 second was averaged and plotted as a function of the step number, demonstrating clear sub-nm spatial increments.”

3. In a second experiment the authors use the same platform to perform SERS on hemoglobin. Figure 3 illustrates some reproducibility tests and reports (fig. 3b) some number for trapping force. Anyway, also here no details are reported. Looking at the experimental spectra it seems that they can just detect few main Raman peaks (with long acquisition time (5 sec) and no so low concentration (5 microM)).

We have re-conducted the SERS characterizations of hemoglobin at lower concentration (100 nM) with shorter acquisition time (1 s) and lower laser power (10%, 2.5 mW), showing more subtle structural features in the new SERS spectra in Fig. 3. We have added detailed descriptions on the experimental procedure and more Raman peak analysis to the main text and the supplementary information. Our focus is on the characterization of the secondary structures of protein at native states under physiological concentration in aqueous solutions, which is different from other single-molecule SERS measurements that may denature the proteins for extremely high SERS enhancements. We used 100 nM hemoglobin in the new experiments, which is much lower than the physiological concentration of hemoglobin in blood (4.7 mM to 5.6 mM [Wood et al. DOI: 10.1016/B978-0-12-818610-7.00013-X]), in order to verify that our tunable SERS platform can preserve the proteins at the native states. The new SERS spectra of 100 nM hemoglobin are shown in Fig.3. The details of the experimental procedures and the force measurement are added in Fig. 9S and 7S, respectively.

Under the visualized manipulation by optical tweezers, the two AgNP-coated beads were approached at incremental distance and under gradually adjusted Raman detection power to optimize the SERS signal of 100 nM hemoglobin in aqueous solution in Fig. 3a and 3b. When the beads distance was smaller than 20 nm and the laser power was larger than 10% (2.5 mW), the subtle spectral features were overwhelmed by the intense characteristic peaks arisen from amorphous carbon (~ 1370 and 1590 cm^{-1}), suggesting the molecular damage. Thus, the SERS detection window was set under the beads distance at 20 nm and the detection power at 10% with 1 s acquisition time. Since the AgNP-coated beads were separated, there was no force generated between them. The subtle Raman features of hemoglobin were clearly observed, which are consistent with literature data. It is showcasing that the dynamic SERS probe on the optical tweezers coupled Raman microscope could offer adjustable SERS enhancements efficiently and conveniently to preserve the protein native states. We have added detailed descriptions to the main text to show the power of our SERS platform.

Under the optimized experiment condition of 10% Raman excitation power and 20 nm beads distance, we recorded the SERS spectra of 100 nM hemoglobin with 1 s acquisition time, as shown in Fig.3. The oxidation marker band ν_4 (symmetric stretching of pyrrole half-ring, ~ 1376 cm^{-1} for ferric heme and ~ 1346 cm^{-1} for ferrous heme) and spin state marker bands ν_2 ($\text{C}_\beta\text{-C}_\beta$ stretching, ~ 1568 cm^{-1} for 6cHS and ~ 1594 cm^{-1} for 6cLS) in SERS spectra were consistent with those in spontaneous Raman spectra, indicating the native state of hemoglobin retained in our dynamic SERS probe. Other characteristic peaks of hemoglobin such as 1640 cm^{-1} (ν_{10} , $\text{C}_\alpha\text{-C}_m$ asymmetric stretching), 1508 cm^{-1} (ν_3 , symmetric stretching of $\text{C}_\alpha\text{-C}_m$), 1440 cm^{-1} (vinyl deformation), 1404 cm^{-1} (ν_{29} , symmetric stretching of pyrrole quarter-ring), 1177 cm^{-1} (ν_{30} , asymmetric stretching of pyrrole half-ring) and 1128 cm^{-1} (ν_{22} , $\text{C}_\alpha\text{-N}$ stretching) were also in good agreement with spontaneous Raman. Whereas, the marker bands corresponding to of 5-coordinated high-spin heme, representing the non-native state arisen from the interaction between the metal surface and the hemoglobin, were not observed. It is clear that the SERS probe created by two AgNP-coated beads could preserve the oxidation state, the coordination number, and the spin state of hemes in hemoglobin, which are closely linked to the native structure and function of hemoglobin. Furthermore, the high similarity among these SERS spectra demonstrates the reproducibility of this SERS detection window. Histogram analysis of peaks intensity at 1592 cm^{-1} , 1376 cm^{-1} , 1640 cm^{-1} , 1568 cm^{-1} , 1440 cm^{-1} , 1404 cm^{-1} , 1346 cm^{-1} and 1177 cm^{-1} were presented in Fig. 3e, Fig. 3f and Fig. 11S. All the RSDs fall within a satisfactory range of 18.5%, demonstrating reproducible and stable SERS measurement provided by our configuration. We have added the complete peaks assignment of hemoglobin in Table. 1S and the reproducibility analysis on 6 more peaks in Fig. 11S.

Revised Fig. 3 and main text on the SERS characterizations of 100 nM hemoglobin:

Figure 3. The spectroscopic characterizations of hemoglobin at its native states. **a** SERS spectra of 100 nM hemoglobin in aqueous solution when the two AgNP-coated beads were

trapped at different distance (50 nm to 10 nm). **b** SERS spectra of 100 nM hemoglobin solution under different Raman excitation power (5 % to 100 %). **c** The comparison between SERS spectra of 100 nM hemoglobin solution (blue) with 1 s acquisition time and spontaneous Raman spectrum of 250 μ M hemoglobin solution with 60 s acquisition time (black). **d** 3D stacking plot of SERS spectra of 100 nM hemoglobin solution obtained from AgNP-coated beads trapped at 20 nm with 1 s acquisition time. **e** and **f** The histograms of the intensities of the hemoglobin characteristic peaks at 1376 cm^{-1} (mean=4929.15, RSD=14.63%) and 1594 cm^{-1} (mean=5391.41, RSD=16.35%) across the 50 SERS spectra in **d**, respectively.”

“We utilized the controllable SERS probe as a detection window to analyze the flowing hemoglobin in dilute aqueous solution without incubation or aggregation with AgNPs. As illustrated in Fig. 9S, two AgNP-coated beads were trapped in the microfluidic bead channel then moved to the hemoglobin channel for SERS measurements. The flow rate of the hemoglobin solution was fine-tuned to minimize the interaction between proteins and AgNP-coated beads. The spectroscopic scans of the individual AgNP-coated beads at the Raman excitation spot show the blank spectra in Fig. 10S to confirm neither intra-bead hotspots existence nor hemoglobin attachments. Under the visualization and manipulation by optical tweezers, the two AgNP-coated beads were approached at incremental distance and gradually adjusted Raman excitation power to optimize the SERS signal of 100 nM hemoglobin in aqueous solution in Fig. 3a and 3b. When the beads distance was smaller than 20 nm and the laser power was larger than 10% (2.5 mW), the subtle spectral features were overwhelmed by the intense characteristic peaks arisen from amorphous carbon ($\sim 1370 \text{ cm}^{-1}$ and $\sim 1580 \text{ cm}^{-1}$)⁵², suggesting molecular damages. Thus, the SERS detection window was set under the bead distance at 20 nm and the Raman excitation power at 10% with 1 s acquisition time. It is showcasing the great adjustability of the optical tweezers-coupled Raman spectroscopy to preserve the protein native states.

To examine if the hemoglobin retains native, five SERS spectra of 100 nM hemoglobin with 1 s acquisition time (blue) were compared to the spontaneous Raman spectrum of 250 μ M hemoglobin with 60 s acquisition time (black) in Fig. 3c. The identical frequencies between the SERS spectra and the Raman spectrum of hemoglobin indicates the similar protein states in measurements³. As the oxidation state marker bands, the subtle peaks at 1346 cm^{-1} and 1376 cm^{-1} among these spectra are distinctly similar, attributed to the ferrous state and the ferric state⁵³, respectively. Vibrational bands at 1568 cm^{-1} and 1594 cm^{-1} corresponding to 6-coordinated high-spin heme and 6-coordinated low-spin heme are apparent at matching positions^{53, 54}. Bands appearing at 1089 cm^{-1} , 1311 cm^{-1} and 1440 cm^{-1} , assigned to the vinyl group deformation in the porphyrin ring of the heme center, are in good agreements.⁵⁵ Whereas, the marker bands of 5-coordinated high-spin heme at 1494 cm^{-1} and 1572 cm^{-1} ^{53, 54} representing the non-native state from the perturbation of metal surface, were not observed.

All peak assignments of hemoglobin are listed in Table. 1S^{53, 55, 56, 57, 58}. It is clear that the SERS probe created by two AgNP-coated beads well preserved the oxidation state, the coordination number, and the spin state of hemes in hemoglobin, which are closely linked to the native structure and function of hemoglobin⁵⁴. Furthermore, the high similarity of these SERS spectra demonstrates the reproducibility of this SERS platform.

To further preserve the protein native states, the two AgNP-coated beads were replaced freshly in parallel SERS measurements to minimize the interaction time with proteins^{19, 59}. Fig. 3d displays SERS spectra of 100 nM hemoglobin solution obtained from 50 parallel measurements. Apparently, the vibrational signatures of the heme center of hemoglobin at 1346, 1376, 1568, and 1594 cm^{-1} among the 50 SERS spectra are approximately the same. Fig. 3e and 3f demonstrate the histograms of the peak intensities at 1376 cm^{-1} and 1594 cm^{-1} across the 50 SERS spectra with relative standard deviation (RSD) as 14.63% and 16.35%, respectively. Histogram analysis of other spectral features is presented in Fig. 11S. These reproducible and stable spectra prove the consistent SERS enhancements in the parallel measurements when two AgNP-coated beads were trapped at a constant distance. Hence, the controllable SERS probe inside the microfluidic flow chamber could preserve the flowing proteins at their native states and generate reproducible SERS spectra.”

Newly added Table 1S for the complete peak assignments of hemoglobin and Fig. 11S for the reproducibility analysis on 6 more peaks:

“Table 1S. The vibrational bands assignment of hemoglobin.”

Raman Shift (cm ⁻¹)	Assignment*
755	ν_{15} , ν (pyr breathing)
1006	ν_{45} , ν (C $_{\alpha}$ -C $_1$) _{asym}
1089	δ (=CH $_2$) _{asym}
1128	ν_{22} , ν (C $_{\alpha}$ -N)
1177	ν_{30} , ν (pyr half-ring) _{asym}
1232	ν_{13} , δ (C $_m$ -H)
1311	δ (CH=)
1346 for Fe(II)	ν_4 , ν (pyr half-ring) _{sym}
1376 for Fe(III)	
1404	ν_{29} , ν (pyr quarter-ring) _{sym}
1440	δ_s (=CH $_2$)
1508 for 6cLS	ν_3 , ν (C $_{\alpha}$ -C $_m$) _{sym}
1568 for 6cHS	
1594 for 6cLS	ν_2 , ν (C $_{\beta}$ -C $_{\beta}$)
1640	ν_{10} , ν (C $_{\alpha}$ -C $_m$) _{asym}

*Assignments are based on the studies by Hu et al.⁹, Kalaivani et al.¹⁰, Wood et al.¹¹, Casella et al.¹², and Mizutani¹³.”

Figure 11S. The histogram of the intensities of the hemoglobin characteristic peaks at 1640 cm⁻¹ (mean=4413.01, RSD=15.50%), 1568 cm⁻¹ (mean=5139.80, RSD=18.23%), 1440 cm⁻¹

(mean=3811.98, RSD=16.13%), 1404 cm^{-1} (mean=4256.20, RSD=15.74%), 1346 cm^{-1} (mean=3876.91, RSD=15.93%) and 1177 cm^{-1} (mean=3095.36, RSD=17.35%) across the 50 SERS spectra in Fig. 3 d, respectively. All the RSDs fall within a range of 18.5%, demonstrating reproducibility and stability of our SERS platform.”

Newly added Fig. 9S to illustrate the experimental procedures:”

Figure 9S. **a**: Illustration of the microfluidic flow cell with three adjacent laminar fluidic streams from sample channel, buffer channel, and AgNP-coated bead channel. **b**: Illustration of the detection procedure: Step 1: Two AgNP-coated beads are trapped at AgNP-coated beads stream. Step 2: Two trapped AgNP-coated beads are placed at sample stream. Step 2.5: Individual trapped AgNP-coated beads is brought to the Raman spot for spectroscopic scan. Step 3: Two AgNP-coated beads are approached to generate SERS enhancement for sample measurements.”

Newly added Fig. 7S to illustrate the force measurement (Since the AgNP-coated beads were separated at 20 nm, there was no force generated between them during the SERS

characterizations of hemoglobin. The force emerged only when the two AgNP-coated beads were approached at 0 nm):

Figure 7S. a, c: Illustrations of the force measurement on the trapped AgNP-coated beads by the optical tweezers and b: time traces of the force detected on the optical tweezers when two AgNP-coated beads were separated and touched. When two trapped AgNP-coated beads were pushed at the bead distance of 0 nm, the force has increased rapidly from 0 pN to 40 pN, indicating the contact of two AgNP-coated beads to ensure the stability of the interparticle hotspot at the maximum SERS enhancement.”

4. In the final experiment the authors want to demonstrate that the proposed platform can be used to investigate on the conformational states of alpha-synuclein at physiological concentration. To me this part is also very critical and MUST be discussed taking into consideration the following points:

The authors claim that it's possible to discriminate alfa-helical, beta-sheet and random-coil conformations. The protein itself, presents sections that are mainly alfa-helical and beta-sheet, as well as random coil. To me, in a SERS experiment where nanogaps between Ag-nanoparticles are used as Hot-Spots, it's not possible to know if the difference in the spectra are due to different section of the protein interacting with the hot-spot or are due to different protein conformations. The size of the hot-spot, in fact, can be very small and comparable to sub-section of the protein, so justifying different spectrum for different measurements. This phenomena is very well discussed in doi.org/10.1002/anie.202000489. In order to demonstrate that the platform is able to detect protein conformation, the authors should consider additional experiments where different proteins (at least 2 with well known conformations) are used.

We thank the reviewer’s suggestion concerning the size of proteins and that of the hotspot, which can showcase the power of our platform in the control of SERS active interparticle gap with high spatial resolution. We have conducted additional experiments on two well-known

globular proteins: lysozyme and bovine serum albumin (BSA) to verify that our SERS platform can detect the whole protein conformations. Moreover, we have added the introduction on the heterogeneous nature of alpha-synuclein as an intrinsically disordered protein comparing to those well-known globular proteins with homogeneous structures to the main text.

Since the coupled optical tweezers on our SERS platform can manipulate the AgNP-coated beads with sub-nanometer spatial resolution, we are able to fine-tune the size of the hotspot to accommodate various analytes, from small chemical compounds to biomacromolecules. To avoid molecular damage, the SERS detection window was set under the beads distance at 20 nm and the detection power at 10% with 1 s acquisition time. Thus, the size of the hotspot is bigger than the size of proteins. For example, the size of the globular lysozyme is 45 Å x 30 Å x 30 Å. (*Nature* **206**, 757–761 (1965).) The SERS spectra of lysozyme is highly reproducible and consistent with its Raman spectrum and the literature SERS spectra (doi: 10.1016/0584-8539(94)00225-Z), representing the conformation of the whole protein. Alpha-synuclein has similar amino acid numbers with lysozyme, which should be smaller than the 20 nm hotspot. Besides, the recommended paper doi.org/10.1002/anie.202000489 focuses on the primary structure of short peptides, i.e. amino acid sequence, sidechain conformation on metal surface, thus the short peptides were absorbed on SERS substrates and characterized in around 5 nm hotspot. While the focus of our manuscript is on the secondary structure of proteins with the signature vibrations in the region of amide I and III. Although our SERS platform can also create very small hotspot with sub-nanometer resolution, we just use a large one to allow the whole proteins passing through freely in the liquid flow to minimize the interaction with metal surface. The SERS spectra of hemoglobin, lysozyme, and BSA indicate the preservation of proteins at the native states and the native conformations in a whole on our platform.

As a typical intrinsically disordered protein, alpha-synuclein lacks stable secondary and tertiary structures in aqueous solution, but could self-assemble to different transient species with various structures in low population and eventually grow into amyloid fibrils. It is challenging to characterize these heterogeneous transient species among the dominating disordered monomeric species, because different species are existing in a dynamic equilibrium. Since our platform enables the sensitive SERS characterization of proteins with reproducible enhancements, we could directly identify the structural variations arisen from the heterogeneous transient species of alpha-synuclein under 200 parallel small-size sampling SERS measurements in dilute concentration (1 μM) and short accumulation time (1 s). Among 200 parallel small-size sampling SERS measurements, the amide I and the amide III bands of the SERS spectra of 1 μM alpha-synuclein exhibit prominent variations in comparison to the uniform spectral features of 1 μM lysozyme, indicating the co-existence of diverse conformations and the direct characterization of the transient species of alpha-synuclein at

physiological concentration. The probability to observe monomers in random coil structures is very high, since they are predominant species. Occasionally, we could detect the transient species showing α -helix or β -sheet structures. Such direct identification of the structural variation of alpha-synuclein demonstrates that our sensitive SERS platform could reduce the ensemble averaging from the bulk measurement to reveal more structural details of its transient species. The small structural fluctuation of lysozyme (containing different structural sections in individual protein but possessing the identical folded structure from one to another) and the large structural fluctuation of alpha-synuclein (existing in different conformations from one to another) represents the homogeneous nature of compact globular proteins and the heterogeneous nature of intrinsically disordered proteins, respectively.

Newly added description about experimental condition optimization:

“Thus, the SERS detection window was set under the bead distance at 20 nm and the Raman excitation power at 10% with 1 s acquisition time. It is showcasing the great adjustability of the optical tweezers-coupled Raman spectroscopy to preserve the protein native states.”

Newly added Fig. 4 for the new SERS measurements of lysozyme:

Figure 4. The spectroscopic characterizations of lysozyme in its compact globular structure. **a** The comparison between 50 SERS spectra of 1 μ M lysozyme solution with 1 s acquisition time (bottom) and the spontaneous Raman spectrum of 1 mM lysozyme solution with 5 min acquisition time (top). **b** Histogram of the Amide I band distribution of the 50 SERS spectra of 1 μ M lysozyme solution in **a**, indicating the mean as 1655 cm⁻¹ with 0.1% RSD. **c** Illustration of the ensemble averaging from the spontaneous Raman measurement of lysozyme in the concentrated solution and the small-size sampling from the SERS measurements of lysozyme in the dilute solution.”

Newly added result and discussion about the SERS measurements of lysozyme and BSA:

“To demonstrate the ability to detect protein conformations with high accuracy, we employed the dynamic SERS probe as a detection window to characterize two well-known globular proteins in solutions: lysozyme and bovine serum albumin (BSA). With the experimental

protocol analogous to the previous section, two fresh AgNP-coated beads were trapped at 20 nm to analyze the flowing proteins in dilute aqueous solutions, which would be replaced freshly in parallel experiments. Fig. 4a displays the SERS spectra of 1 μM lysozyme solution acquired from 50 parallel experiments, showing the vibrational frequencies of lysozyme identical to its spontaneous Raman spectrum. Specifically, the peaks at 766, 1015, 1337, 1557 cm^{-1} are assigned to the aromatic residues (tryptophan) and the peak at 1450 cm^{-1} is attributed to aliphatic residues (CH_2)⁶⁰. The amide I band at 1655 cm^{-1} and the amide III band at 1250 cm^{-1} imply the existence of α -helix in the folded globular structure of lysozyme. The spectral contributions from different secondary structures were deconvolved to α -helix (45.2%), β -sheet (11.3%) and random coil (43.5%) in Fig. 12S, which is consistent with the previous investigations^{61, 62}. Fig. 4b demonstrates the Amide I band distribution of the 50 SERS spectra of 1 μM lysozyme solution with $1655 \pm 2 \text{ cm}^{-1}$ (0.1% RSD), indicating its structural stability and homogeneity as a typical globular protein. Moreover, the SERS spectra of 1 μM BSA in Fig. 13S and Fig. 14S also provide the structural component assessment (66.5% α -helix, 9.5% β -sheet and 24.1% random coil) supported by the previous studies⁶³. It is worth noting that the 50 SERS spectra of 1 μM lysozyme in Fig. 4a demonstrate high similarities, due to the stability and reproducibility of this SERS probe as well as the nature of the stable, compact, and globular conformation of lysozyme. Since the sizes of these globular proteins are smaller than the bead distance at 20 nm⁶², these SERS spectra reflect the whole protein structures. As illustrated in Fig. 4c, the small-size sampling in the parallel SERS measurements unveil the protein structural fluctuation to complement the ensemble averaging in the spontaneous Raman measurement.”

Revised Fig. 5 for the new SERS measurements of alpha-synuclein:

Figure 5. The spectroscopic characterizations of an intrinsically disordered protein: alpha-synuclein. **a** CD spectrum of 200 μM alpha-synuclein in aqueous solution. **b** Spontaneous

Raman spectra of 2 mM (green) and 250 μ M (blue) alpha-synuclein solution with 10 min acquisition time. **c** The comparison among three representative types of SERS spectra of 1 μ M alpha-synuclein solution with 1 s acquisition time (blue, red, and black) and the SERS spectrum of 250 μ M alpha-synuclein solution with 5 min acquisition time (purple). **d** Mapping of 200 SERS spectra of 1 μ M alpha-synuclein solution obtained from two AgNP-coated beads trapped at 20 nm with 1 s acquisition time. The color bar shows the normalized intensities from low (dark blue) to high (red). **e** Illustration of the ensemble averaging from the measurement of alpha-synuclein at high concentration with long accumulation time and the small-size sampling from the measurements of alpha-synuclein at low concentration with short accumulation time.”

Revised result and discussion about the new SERS measurements of alpha-synuclein:

“The physiological concentration of alpha-synuclein at non-aggregated states is 1 μ M⁴⁷, below the detection threshold of Raman spectroscopy. Whereas the experimental limit of detection (LOD) of alpha-synuclein on our SERS platform is 100 nM, thus this sensitive SERS approach is feasible to characterize the transient species of alpha-synuclein in dilute solutions. (Details are shown in Fig. 15S and Fig. 20S.) Similar to the experimental protocol in previous section, two fresh AgNP-coated beads were trapped at 20 nm as the dynamic SERS window to characterize the flowing alpha-synuclein on our platform. Fig. 5c demonstrates three representative types of SERS spectra of 1 μ M alpha-synuclein solution with 1 s acquisition time among the 200 parallel experiments shown in Fig. 5d. Strikingly, the amide I and the amide III bands of the SERS spectra of 1 μ M alpha-synuclein exhibit prominent variations in comparison to the uniform spectral patterns of 1 μ M lysozyme in Fig. 4, indicating the co-existence of different alpha-synuclein species with various structures at physiological concentration⁷. In Fig. 5c, the spectral characteristics of the blue SERS spectrum of 1 μ M alpha-synuclein fall in the intervals of 1650-1657 cm^{-1} (amide I bands) and 1270-1300 cm^{-1} (amide III bands), indicating the alpha-synuclein species in α -helix structure^{16, 65}. The red SERS spectrum of 1 μ M alpha-synuclein shows the amide I band at around 1662-1665 cm^{-1} and the amide III band at 1230-1240 cm^{-1} , which are associated with β -sheet structure^{16, 17, 65}. This assignment is further confirmed by the SERS spectrum of 1 μ M alpha-synuclein at pH 3 in Fig. 16S, since alpha-synuclein folds into ordered β -sheet conformations in acidic conditions. The amide I band at around 1671 cm^{-1} and the amide III band at 1240-1250 cm^{-1} from the black SERS spectrum of 1 μ M alpha-synuclein are attributed to random coil structure^{16, 65, 66}, consistent with its spontaneous Raman spectrum. While the vibrational fingerprints from Phe (1006 cm^{-1}) and deformation from aliphatic residues CH₂ and CH₃ (1450 cm^{-1}) of alpha-synuclein¹⁶ are still uniform across these SERS spectra of 1 μ M alpha-synuclein, since they are insensitive to the change of protein conformations. All the peak assignments of the 1 μ M alpha-synuclein SERS spectra are summarized in Table 2S¹⁶, since the subtle spectral features of alpha-

synuclein acquired at physiological concentration could reveal the structural details of its transient species with great biological significance. In particular, the direct characterization of the β -sheet containing oligomers among the unstructured monomers of alpha-synuclein might provide new insight to the pathological aggregation of alpha-synuclein at the very early stage as this conformation is involved prior to fibrillation^{67, 68}.

Owing to high sensitivity and stability, our SERS platform resolved the structural variations of alpha-synuclein arisen from its transient species, which is the first SERS characterization of alpha-synuclein at physiological concentration.”

5. In conclusion, the idea of 2-laser optical tweezers and controlled SERS is very good and promising, but the manuscript, in order to be published in a high impact journal must be better presented (also the quality of the figures are really bad!). A deeper discussion on the tweezing system to "sub-nm" control the gap and additional experiments on sensitivity are needed.

We highly appreciate the reviewer's valuable suggestions. We have re-conducted all the experiments and added new experiments to improve the data quality. The relating figures (Fig. 2, 3, 4, and 5) and texts (main content and supplementary information) were completely revised and rewritten. In particular, we have performed new complementary experiments to verify the sub-nanometer spatial resolution for the control of the AgNP-coated beads by the optical tweezers on our platform. These results demonstrate that the AgNP-coated bead can reach the designated distance after each optical tweezer movement, even with the step size as small as 0.5 nm. Moreover, we have conducted new experiments and simulations to verify that our optical tweezers-coupled SERS platform is able to reach single-molecule level sensitivity. The characteristic peaks of Rhodamine B are observed even at 10^{-11} M, under the bead distance at 0 nm and the excitation power at 25 mW. According to the fourth power-approximation of electromagnetic enhancement factor (EF) in 3D-FDTD simulation, the maximum EF is as high as 10^9 , which is sufficient to empower the single-molecule level detection. With the precise control of the AgNP-coated beads, our SERS platform can achieve not only the high SERS enhancements but also the tunable, reversible, and reproducible SERS enhancements, which make it suitable for the characterization of various analysts, from small chemical compounds to biomacromolecules. More importantly, the focus of our manuscript is to show the application of SERS on characterizing the secondary structure of proteins at native states in dilute aqueous solutions, especially the structural variation of intrinsically disordered proteins arisen from the transient species with different structures at physiological concentration.

Reviewers' Comments:

Reviewer #1:

Remarks to the Author:

The authors answered all of my questions, satisfactorily. The quality of the manuscript has been increased. Now it is suitable for publication, but first, we need to adapt the format of the references to the Nature Comm. guidelines (e.g., journal names).

Reviewer #2:

Remarks to the Author:

I thank the authors for taking into account the quantity of comments I had made and to reply to them point by point in a precise way. I really appreciated the quantity of new experiments they have made and the improvement of the results in terms of reproducibility, understanding of the process and quality of the spectra.

Also the clarity with which experiments are described has improved, in my opinion.

I think the results are now much more sound and suggest that, configurations capable to manipulate and approach in a controlled way plasmonic particles, open the doors to precise SERS sensing of molecules dispersed in liquid at concentrations not imaginable by conventional Raman.

I suggest publication in the present form

Reviewer #3:

Remarks to the Author:

The updated version of the manuscript is significantly improved with respect to the original version.

The authors did a lot of work in order to reply to all the comments received.

Anyway, to me, there is still a major point not clear in the manuscript. The authors claim nm (or sub-nm) resolution in the nanobeads trapping and gap formation. In the S.I. they now show some plots on force and distance measurements. I know that in literature such spatial control was demonstrated, and it's reasonable that the authors can get it here considering the SERS spectra reported. Anyway, in order to accept the paper for publication in Nat. Comm. I think that it's important to better describe how these measurements have been done (in a convincing way).

Reviewer #1 (Remarks to the Author):

The authors answered all of my questions, satisfactorily. The quality of the manuscript has been increased. Now it is suitable for publication, but first, we need to adapt the format of the references to the Nature Comm. guidelines (e.g., journal names).

We appreciate the reviewer's recommendation. We have revised the format of the reference according to the guidelines of Nature Communications. In particular, we have changed the journal names into abbreviations.

Reviewer #2 (Remarks to the Author):

I thank the authors for taking into account the quantity of comments I had made and to reply to them point by point in a precise way. I really appreciated the quantity of new experiments they have made and the improvement of the results in terms of reproducibility, understanding of the process and quality of the spectra.

Also the clarity with which experiments are described has improved, in my opinion.

I think the results are now much more sound and suggest that, configurations capable to manipulate and approach in a controlled way plasmonic particles, open the doors to precise SERS sensing of molecules dispersed in liquid at concentrations not imaginable by conventional Raman.

I suggest publication in the present form

We appreciate the reviewer's recommendation. The reviewer's suggestions and comments are very helpful for us to improve this research. We hope to expand the scope of this research for wider applications in the future.

Reviewer #3 (Remarks to the Author):

The updated version of the manuscript is significantly improved with respect to the original version. The authors did a lot of work in order to reply to all the comments received.

Anyway, to me, there is still a major point not clear in the manuscript. The authors claim nm (or sub-nm) resolution in the nanobeads trapping and gap formation. In the S.I. they now show some plots on force and distance measurements. I know that in literature such spatial control was

demonstrated, and it's reasonable that the authors can get it here considering the SERS spectra reported. Anyway, in order to accept the paper for publication in Nat. Comm. I think that it's important to better describe how these measurements have been done (in a convincing way).

We appreciate the reviewer's suggestion and recommendation. We have added the detailed description of these measurements in Fig. 7S, referring to doi:10.1073/pnas.0603342103 and doi: 10.1038/ncomms6885. In addition to the literature support, we are able to use optical tweezers to move two beads apart in 0.5 nm increments while the distance is recorded precisely from 0.0 nm to 2.5 nm in Fig. 4S. Thus, both literatures and our results support the claim of the sub-nanometer spatial resolution on our optical tweezers-coupled Raman platform.

Revised Fig. 7S with the description of force and distance measurements:

Figure 7S. a and e: The real-time camera images of two AgNP-coated beads separated and touched, respectively. **b and f:** Illustrations of the force and distance measurements on the two AgNP-coated beads trapped by optical tweezers when beads are separated and touched. The magnitude of force with respect to the displacement of the bead from the center of the trap follows Hooke's law, $\mathbf{F} = -k \cdot \mathbf{x}$, where k is the trap stiffness and \mathbf{x} is the bead displacement detected by a position sensitive detector (PSD). The distance between two trapped beads is measured by the camera-based image tracking and the trap position corrected by the bead displacement. **c:** Time traces of the detected force when two AgNP-coated beads were separated and touched. When two trapped AgNP-coated beads were pushed at the bead distance of 0 nm, the force increased sharply from 0 pN to 40 pN, indicating the contact of two AgNP-coated beads to ensure the stability of the interparticle hotspot at the maximum SERS enhancement. **d:** The statistical analysis of the data fluctuation in the first 5 s, giving mean = -0.002 pN and SD < 0.1 pN as the force resolution (SD < 0.3 nm as the spatial resolution at the trap stiffness $k = 0.3$ pN/nm).”

Revised Fig. 4S with the description of the measurement of the distance against the bead incremental movement:

Figure 4S. Incremental movements of two AgNP-coated beads by optical tweezers at different step sizes. **a**: Illustration of the incremental bead movements. One bead (on the left side) was kept stationary while the other bead (on the right side) was moved away in the incremental steps (5 steps in total). After each steering operation, the distance data points of the two trapped AgNP-coated beads within 1 second was averaged and plotted as a function of the step number with the real-time camera images in **b**: at 3.0 nm step size, **c**: at 2.0 nm step size, **d**: at 1.0 nm step size, and **e**: at 0.5 nm step size, respectively, demonstrating the sub-nm spatial resolution in the bead trapping and the gap formation on our optical tweezers-coupled Raman platform.”